# 2,4-dienoyl-CoA reductase regulates lipid homeostasis in treatment-resistant prostate cancer

Arnaud Blomme [1], Catriona A. Ford[1], Ernest Mui[2], Rachana Patel[1], Chara Ntala[1,2], Lauren E. Jamieson [3], Mélanie Planque[4,5], Grace H. McGregor[1,2], Paul Peixoto [6,7,8], Eric Hervouet [6,7,8], Colin Nixon[1], Mark Salji[2], Luke Gaughan[9], Elke Markert[2], Peter Repiscak[1], David Sumpton [1], Giovanny Rodriguez Blanco [1], Sergio Lilla [1], Jurre J. Kamphorst [1,2], Duncan Graham [3], Karen Faulds[3], Gillian M. MacKay[1], Sarah-Maria Fendt [4,5], Sara Zanivan [1,2] & Hing Y. Leung [1,2✉]

Despite the clinical success of Androgen Receptor (AR)-targeted therapies, reactivation of AR signalling remains the main driver of castration-resistant prostate cancer (CRPC) progression. In this study, we perform a comprehensive unbiased characterisation of LNCaP cells chronically exposed to multiple AR inhibitors (ARI). Combined proteomics and metabolomics analyses implicate an acquired metabolic phenotype common in ARI-resistant cells and associated with perturbed glucose and lipid metabolism. To exploit this phenotype, we delineate a subset of proteins consistently associated with ARI resistance and highlight mitochondrial 2,4-dienoyl-CoA reductase (DECR1), an auxiliary enzyme of beta-oxidation, as a clinically relevant biomarker for CRPC. Mechanistically, DECR1 participates in redox homeostasis by controlling the balance between saturated and unsaturated phospholipids. *DECR1* knockout induces ER stress and sensitises CRPC cells to ferroptosis. In vivo, *DECR1* deletion impairs lipid metabolism and reduces CRPC tumour growth, emphasizing the importance of DECR1 in the development of treatment resistance.

[1] CRUK Beatson Institute, Garscube Estate, Switchback Road, Glasgow G61 1BD, UK. [2] Institute of Cancer Sciences, University of Glasgow, Garscube Estate, Switchback Road, Glasgow G61 1QH, UK. [3] Centre for Molecular Nanometrology, Department of Pure and Applied Chemistry, Technology and Innovation Centre, University of Strathclyde, 99 George Street, Glasgow G1 1RD, UK. [4] Laboratory of Cellular Metabolism and Metabolic Regulation, VIB-KU Leuven Center for Cancer Biology, VIB, Herestraat 49, 3000 Leuven, Belgium. [5] Laboratory of Cellular Metabolism and Metabolic Regulation, Department of Oncology, KU Leuven and Leuven Cancer Institute (LKI), Herestraat 49, 3000 Leuven, Belgium. [6] Univ. Bourgogne Franche-Comté, INSERM, EFS BFC, UMR1098, Interactions Hôte-Greffon-Tumeur/Ingénierie Cellulaire et Génique, 25000 Besançon, France. [7] EPIGENExp (EPIgenetics and GENe EXPression Technical Platform), Besançon, France. [8] DIMACELL Dispositif Interrégional d'Imagerie Cellulaire, Dijon, France. [9] Northern Institute for Cancer Research, The Medical School, Newcastle University, Framlington Place, Newcastle upon Tyne NE2 4HH, UK. ✉email: h.leung@beatson.gla.ac.uk

Prostate cancer is the most common cancer amongst men in the western world[1]. Due to its reliance on androgen receptor (AR) signalling, androgen deprivation therapy (ADT) has long been the standard of care for advanced disease. However, although the majority of patients initially respond to hormonal therapy, they ultimately relapse and progress to a lethal form of the disease, termed castration-resistant prostate cancer (CRPC).

Over the last decades, specific AR inhibitors (ARI) have shown promising effects in the clinic, significantly improving patient outcomes[2]. Bicalutamide, a first-generation AR antagonist, has been successfully used for almost two decades[3] before being progressively replaced by enzalutamide (MDV3100), which displayed better in vitro characteristics[4] and improved efficacy in patients[5,6]. In phase 3 clinical trials, enzalutamide treatment significantly improved progression-free survival in metastatic[7,8], and non-metastatic CRPC patients[9]. Apalutamide (ARN-509) is another novel AR antagonist that recently completed Phase 3 clinical evaluation, significantly reducing the risk of metastasis formation in men with high risk non-metastatic CRPC[10].

Despite initial response to enzalutamide and apalutamide, AR mutations[11–14], gene amplification[15], aberrant splicing[16] or signalling bypass[17] can all account for resistance to AR-targeted therapies. Therefore, a better understanding of the adaptive tumour phenotype following treatment resistance will help to identify novel therapeutic approaches to tackle AR-proficient CRPC.

AR signalling critically regulates cellular metabolism in prostate cancer[18]. Hence, targeting metabolism represents an appealing option to overcome resistance to AR-targeted therapies. In comparison to other cancers, prostate cancer displays very specific metabolic features[19] such as an early reliance on mitochondrial metabolism rather than glycolysis, although the latter becomes important as the disease progresses[20]. Prostate cancer is also characterised by profound alterations in cholesterol and lipid metabolism[21], highlighted by dysregulation of both fatty acid synthesis and oxidation pathways. This rewiring of lipid metabolism offers new therapeutic opportunities and has led to the development of multiple inhibitors, some of which are currently undergoing clinical trials.

In this study, we use a combination of proteomics and metabolomics to perform an unbiased characterisation of LNCaP-derived cell lines chronically exposed to long-term bicalutamide, apalutamide or enzalutamide treatment. We show that long-term resistance to AR inhibition is sustained by profound changes in glucose and lipid metabolism. This metabolic rearrangement is mainly dependent on aberrant AR signalling. In addition, we identify a protein signature associated with acquired resistance to ARI. Among the top candidates, 2,4-dienoyl-CoA reductase (DECR1), a mitochondrial enzyme involved in polyunsaturated fatty acid (PUFA) degradation, represents a potential therapeutic target for CRPC. DECR1 deletion in CRPC cells reduces in vitro proliferation and impairs CRPC tumour growth. Mechanistically, we show that DECR1-deficient prostate cancer cells accumulate higher levels of polyunsaturated lipids. This results in a strong ER stress response and an increased sensitivity to GPX4 inhibition, and suggests a potential role for DECR1 in the control of redox homeostasis.

## Results

### ARI-resistant cells display altered but active AR signalling.
To study resistance to AR inhibition, we characterised CRPC derivatives of LNCaP cells that were chronically cultured in the presence of three distinct AR inhibitors (ARI), namely bicalutamide (first generation ARI), apalutamide and enzalutamide

(second generation ARI). When compared to parental LNCaP cells, ARI-resistant cells were larger, and exhibited enhanced cell–cell contact. ARI-resistant cells also generated larger organoid structures when cultured in 3D matrix (Fig. 1a–c). In contrast, ARI-resistant cells proliferated at a slower rate than WT LNCaP (by ~30, 40 and 50% at 72 h for bicalutamide, enzalutamide and apalutamide resistant cells, respectively, Fig. 1d). Similar to what is observed in patients, ARI-resistant cells displayed cross-resistance among the different inhibitors (Fig. 1e).

Because AR is the direct target of the three inhibitors, we first sought to evaluate the status of AR signalling in the ARI-resistant cells. In all three cell lines, expression of full-length AR was maintained or even increased at both protein and mRNA levels (Fig. 1f, g). Expression of the ARv7 variant was also assessed. Bicalutamide-resistant cells displayed a significant increase in expression of the variant, while apalutamide- and enzalutamide-resistant cells showed no change or a decrease in the level of ARv7 (Supplementary Fig. 1a). We then looked at the expression of two canonical target genes downstream of AR, namely KLK3 and FKBP5. While the expression of KLK3 was strongly reduced in all the resistant cells, FKBP5 expression was either not affected or even increased in enzalutamide-resistant cells (Fig. 1f, g). Interestingly, we observed that AR remained in the nucleus of ARI-resistant cells (Fig. 1h, Supplementary Fig. 1b), and was still able to bind the promoter of both target genes (Supplementary Fig. 1c). Altogether these results suggest AR reprogramming, rather than inactivation, as a general survival strategy in ARI-resistant prostate cancer cells.

### ARI resistance is associated with changes in cell metabolism.
To decipher the mechanisms associated with treatment resistance, we characterised the proteome of the parental LNCaP and ARI-resistant cells, cultured in both 2D and 3D conditions (Supplementary Data 1, Supplementary Fig. 2a, b). In comparison to 2D culture, culturing cancer cells in a 3D matrix has been proposed to better mimic the in vivo situation[22] and therefore is of interest in the context of biomarker discovery. We next performed pathway enrichment analysis to identify pathways commonly dysregulated in ARI-resistant cells in 2D and 3D cultures. Interestingly, many metabolic pathways were consistently upregulated in ARI-resistant cells regardless of the culture conditions (Fig. 2a). Indeed, the top upregulated pathways (FDR < 0.05) within each comparison included amino acid, fatty acid, glucose, and glutathione metabolism (Fig. 2a).

Along with exosomal and vesicular proteins, we found that many proteins upregulated in ARI-resistant cells were localised in mitochondria (Supplementary Fig. 2c), suggesting that mitochondrial metabolism might be a key pathway associated with treatment resistance. Mitochondria are at the crossroad between glucose and fatty acid metabolism through the production of citrate and acetyl-CoA. In line with this hypothesis, ARI-resistant cells also displayed increased expression of PPARG and PGC1A, two transcription factors that are crucial for mitochondrial biogenesis and fatty acid metabolism, and of both isoforms of the Liver X receptor (NR1H2 and NR1H3), another critical factor involved in the regulation of lipid homeostasis[23] (Supplementary Fig. 2d).

### Resistant cells exhibit rewired glucose and lipid metabolism.
To better understand the metabolic differences between ARI-resistant and the parental LNCaP cells, we profiled 66 metabolites and identified 16 molecules that were significantly modulated in at least two of the three resistant cell lines (Fig. 2b, FC = 1.5). Intracellular glucose levels were consistently lower in resistant cells, which contrasted with elevated levels of several glycolytic

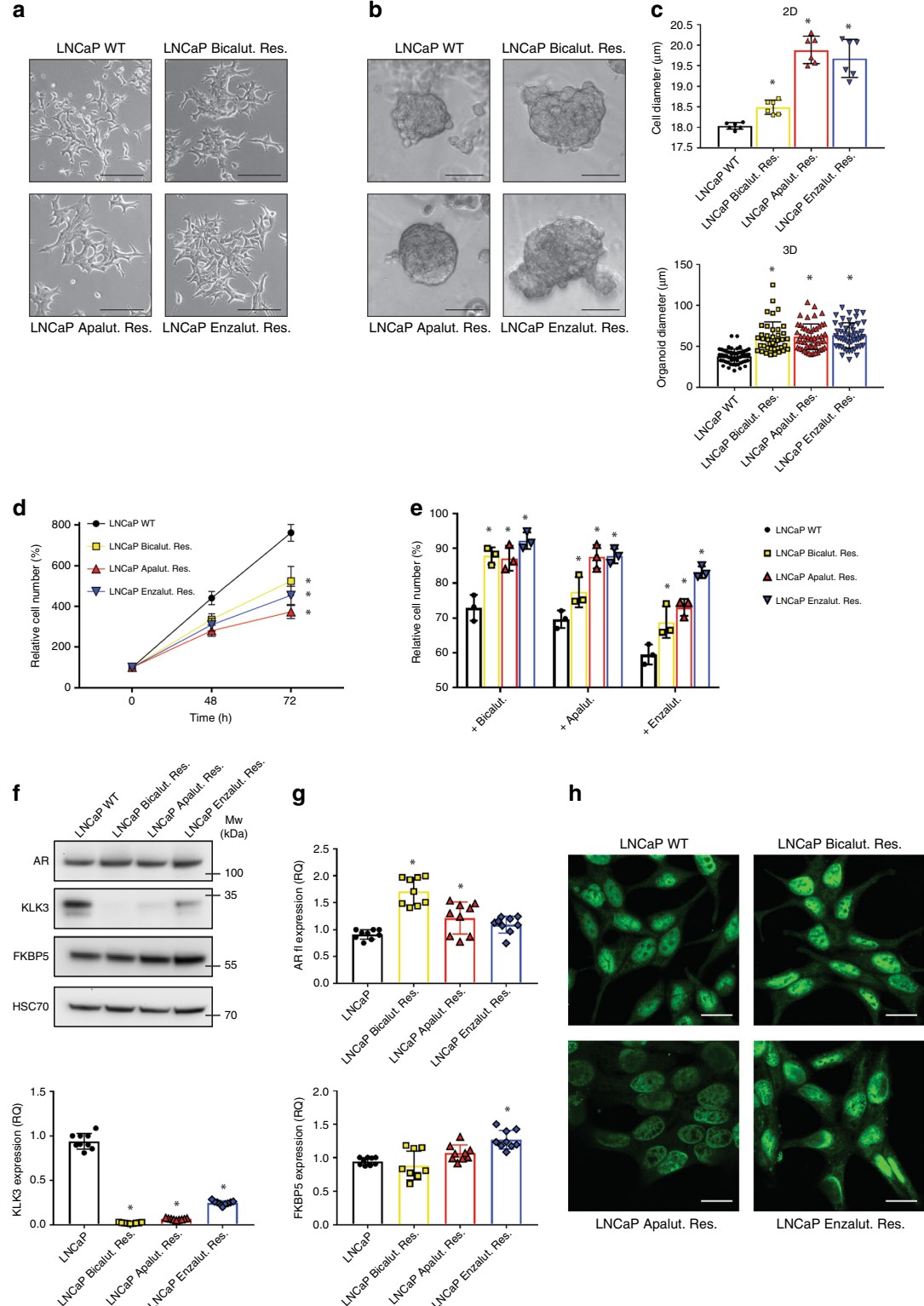

**Fig. 1 AR signalling is conserved in ARI-resistant cells. a** Representative pictures of WT and ARI-resistant LNCaP cells cultured in 2D conditions. Scale bar represents 100 μm. **b** Representative pictures of WT and ARI-resistant LNCaP organoids embedded in Matrigel. Scale bar represents 50 μm. **c** Quantification of cell (top panel) and organoid (bottom panel) diameter. **d** Cell proliferation of WT and ARI-resistant LNCaP cells after 48 and 72 h. Cell count is normalised to initial number of cells at T0. **e** Cell proliferation of WT and ARI-resistant LNCaP cells treated for 48 h with different AR inhibitors (10 μM). Cell count is normalised to non-treated condition. **f** Western blot analysis of AR, KLK3 and FKBP5 expression in WT and ARI-resistant LNCaP cells. HSC70 was used as a sample loading control. **g** RT-qPCR analysis of *AR* (full length, fl), *KLK3* and *FKBP5* expression in WT and ARI-resistant LNCaP cells. *CASC3* was used as a normalising control. **h** Immunofluorescence showing nuclear AR expression in WT and ARI-resistant LNCaP cells. Scale bar represents 20 μm. **c–e**, **g** Data are presented as mean values +/− SD. **c**, **g** *$p$-value < 0.05 using a 1-way ANOVA with a Dunnett's multiple comparisons test. **d**, **e** *$p$-value < 0.05 using a 2-way ANOVA with a Tukey's multiple comparisons test. Source data are provided as a Source Data File.

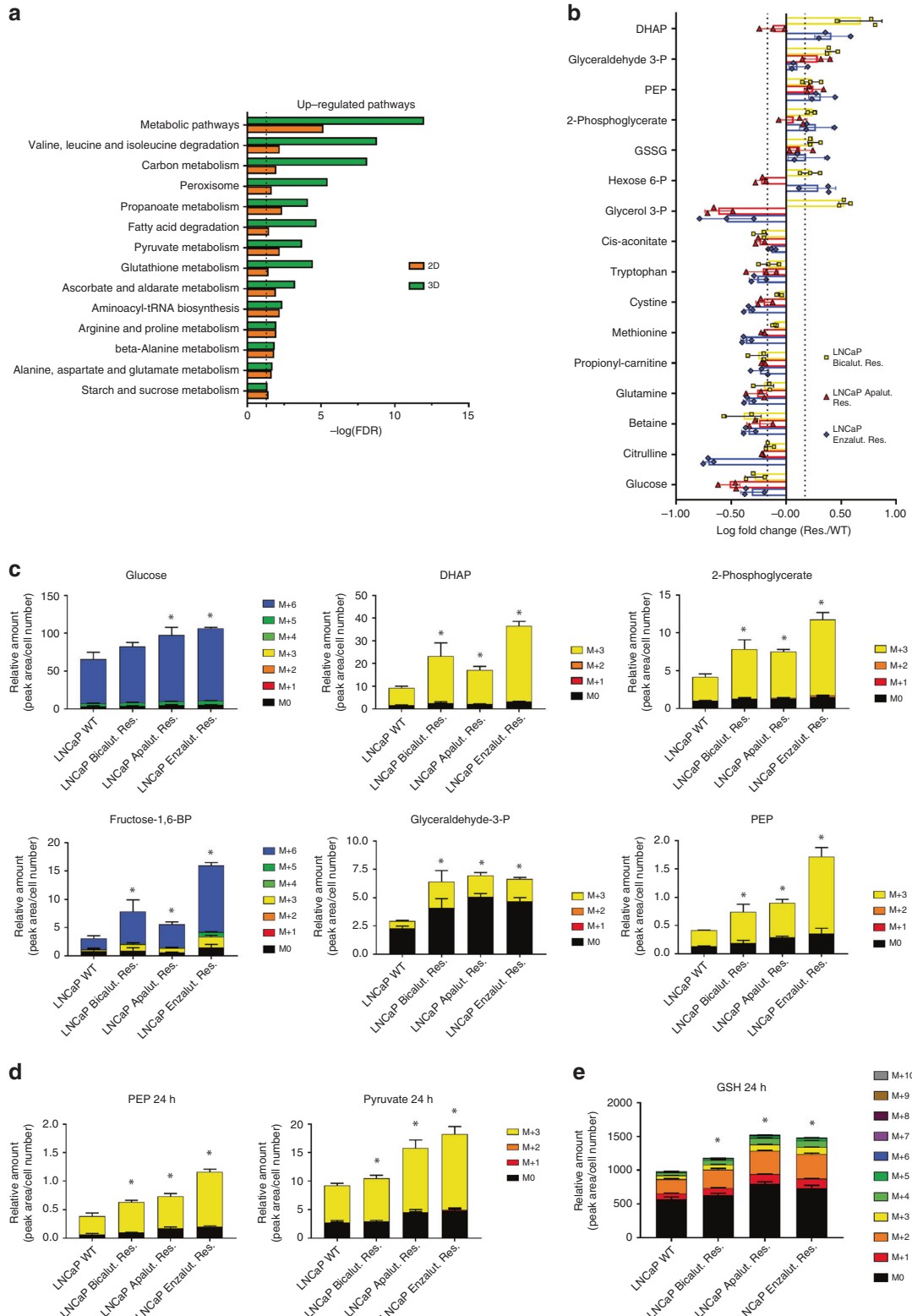

intermediates (such as H6P, G3P, DHAP, 2-PG, and PEP). ARI-resistant cells further displayed elevated expression of the glucose transporter GLUT1 (Supplementary Fig. 2e). Moreover, inhibition of glycolysis using 2-deoxyglucose (2-DG) potentiated the effect of enzalutamide in LNCaP cells (Supplementary Fig. 2f). Combined with our proteomic data (Fig. 2a), these results suggest that ARI-resistant cells may undergo a metabolic shift towards increased glucose consumption. To test this hypothesis, we

cultured the cells for a short period (1 h) in the presence of [U$^{13}$C]-glucose and measured the levels of $^{13}$C-derived metabolites by mass spectrometry. In line with increased glucose metabolism, substantial enrichment of labelled glycolytic intermediates was observed in ARI-resistant cells in comparison to WT LNCaP (Fig. 2c). Levels of labelled pyruvate intermediates remained elevated in ARI resistant cells at a later timepoint (24 h), confirming that glycolysis was more active in the resistant cells

**Fig. 2 Omics analysis of ARI-resistant cells reveal altered glucose metabolism. a** Enriched pathways upregulated in the proteomic analysis of ARI-resistant cells when compared to WT LNCaP. Selected proteins were significantly modulated in at least 2 out of 3 conditions. Pathway enrichment analysis was performed using the STRING database (http://string-db.org). **b** Steady-state levels of significantly regulated metabolites in ARI-resistant cells when compared to WT LNCaP (FC > 1.5, $p < 0.05$ using a 1-way ANOVA with a Dunnett's multiple comparisons test). Selected metabolites were significantly modulated in at least 2 out of 3 conditions. **c** Isotopologue distribution of selected metabolites in ARI-resistant and WT LNCaP cells following $^{13}$C-glucose incorporation for 1 h. **d** Isotopologue distribution of pyruvate and phosphoenolpyruvate in ARI-resistant and WT LNCaP cells following $^{13}$C-glucose incorporation for 24 h. **e** Isotopologue distribution of reduced glutathione (GSH) in ARI-resistant and WT LNCaP cells following $^{13}$C-glucose incorporation for 24 h. **b–e** Data are presented as mean values +/− SD. **c–e** *$p$-value of labelled fraction < 0.05 using a 1-way ANOVA with a Dunnett's multiple comparisons test. DHAP dihydroxyacetone-phosphate, PEP phosphoenolpyruvate, GSH reduced glutathione, GSSG oxidised glutathione. Source data are provided as a Source data File.

(Fig. 2d). Interestingly, a significant proportion of the labelled glucose also served for the generation of glutathione, potentially pointing at the importance of redox homeostasis in resistant cells (Fig. 2a, e).

Glycolysis ultimately produces lactate and pyruvate. Pyruvate can in turn be used in the mitochondria to generate acetyl-coA and support de novo fatty acid synthesis, a process typically dysregulated in aggressive prostate cancer[21]. Therefore, we incubated cells in [U$^{13}$C]-glucose for 72 h and traced $^{13}$C incorporation into palmitic, oleic and stearic acid (Fig. 3a and Supplementary Fig. 3). In all three ARI-resistant cell lines, $^{13}$C enrichment of palmitate, oleate and stearate was significantly higher than in WT LNCaP cells, indicating increased glucose-derived de novo fatty acid synthesis in ARI-resistant cells. In addition, all three resistant lines demonstrated high levels of ACC and ACLY, two regulatory enzymes of the fatty acid synthesis (Fig. 3b). To test whether increased levels of glucose-derived fatty acid synthesis resulted in lipid accumulation in the resistant cells, we compared the cellular lipid profile of the ARI-resistant cells and the parental LNCaP. Strikingly, resistance to AR inhibition was associated with a profound remodelling of the cellular lipidome. We identified 114 lipid molecules that were significantly modulated in all three resistant cell lines, with the majority of them being upregulated in ARI-resistant cells (Fig. 3c, FC = 1.5). All three resistant cell lines displayed a strong and consistent accumulation of numerous species of triglycerides (TG), especially polyunsaturated TG, and sphingolipids (Fig. 3c, right panel). Multiple ceramides and cardiolipin derivatives were also enriched in the resistant cell lines, while several classes of phospholipids, such as phosphatidylcholine (PC) and phosphatidylethanolamine (PE) derivatives, were in contrast over-represented in WT LNCaP (Fig. 3c, left panel).

Altogether, these results support the idea that increased consumption of glucose in the ARI-resistant cells serves, at least in part, to support glutathione and de novo fatty acid synthesis. This metabolic rewiring ultimately leads to lipid accumulation and major changes in the cellular lipidome of the ARI-resistant cells.

**AR drives metabolic reprogramming in ARI-resistant cells**. AMPK is an important regulator of glucose metabolism, and targeting AMPK-dependent autophagy has been proposed to enhance enzalutamide sensitivity[24]. Therefore, we tested whether AMPK activation could contribute to the metabolic rewiring observed in ARI-resistant cells. ARI-resistant cells displayed increased AMPKα expression and activity, as evidenced by increased ACC phosphorylation, in comparison to WT LNCaP cells (Supplementary Fig. 4a). However, silencing of AMPKα expression (Supplementary Fig. 4b) only marginally reduced the levels of labelled glycolytic intermediates generated from $^{13}$C-glucose (Supplementary Fig. 4c).

AR has itself been implicated as a key regulator of glucose metabolism in prostate cancer cells[18]. Therefore, AR could be

responsible for the increased glucose flux observed in ARI-resistant cells. Intracellular glucose levels remained unchanged upon AR depletion in the different cell lines (Supplementary Fig. 4d). In contrast, knockdown of the receptor strongly impaired glucose utilisation in resistant cells, as illustrated by reduced amounts of labelled isotopes of numerous glycolytic intermediates (Fig. 4a). Overall, this effect was much less pronounced in wild type LNCaP cells. In addition, AR-depleted cells displayed reduced expression of ACC and ACLY (Fig. 4b), reinforcing the idea that AR is reprogrammed to promote fatty acid synthesis from glucose in resistant cells.

**Proteomic signature associated with resistance to ARI**. Due to the major metabolic reprogramming observed in ARI-resistant cells, we hypothesised that some initially dispensable enzymes could become essential in maintaining the resistant phenotype. Such candidates would represent interesting targets/biomarkers of ARI resistance. Therefore, we used our proteomic dataset to identify proteins that were consistently modulated in all three ARI-resistant cell lines, in both 2D and 3D conditions. We generated a proteomic panel consisting of 13 candidates that were differentially regulated upon resistance to AR inhibition, independent of the type of inhibitor or the culture conditions (Fig. 5a, Table 1). Identification of well-established factors associated with CRPC such as prostate specific antigen (PSA) or prostate specific membrane antigen (PSMA/FOLH1) strengthened the validity of the proteins identified. Of note, we identified several candidates, mainly enzymes involved in redox processes, which have not yet been reported in prostate cancer (Table 1).

Among these proteins, the mitochondrial 2,4-dienoyl-CoA reductase DECR1 was consistently upregulated in ARI-resistant cells when compared to WT LNCaP. DECR1 is an auxiliary enzyme involved in the degradation of polyunsaturated fatty acid and might therefore be important to support the rewired lipid metabolism associated with ARI resistance (Fig. 3). We first confirmed that DECR1 was overexpressed in ARI-resistant cells (Fig. 5b). Interestingly, acute treatment of multiple AR-proficient prostate cancer cell lines with any of the AR inhibitors was sufficient to modulate DECR1 expression (Fig. 5c) and transient silencing of DECR1 potentiated the effect of enzalutamide in C4-2 cells, a CRPC derivative of LNCaP (Fig. 5d). Conversely, transient overexpression of DECR1 marginally increased cell proliferation in LNCaP cells cultured in the presence of enzalutamide (Fig. 5e).

Potential AR-binding sites have been reported in the *DECR1* gene[25], therefore we sought to test the ability of AR to directly regulate *DECR1*. AR silencing did not affect DECR1 protein levels in any of the LNCaP derivatives (Supplementary Fig. 5a). Similarly, while short-term androgen deprivation induced DECR1 expression in LNCaP, treatment with dihydrotestosterone (DHT) did not show any additional effect (Supplementary Fig. 5b). Finally, AR binding at the promoter of *DECR1* was low in both WT and ARI-resistant LNCaP and was not

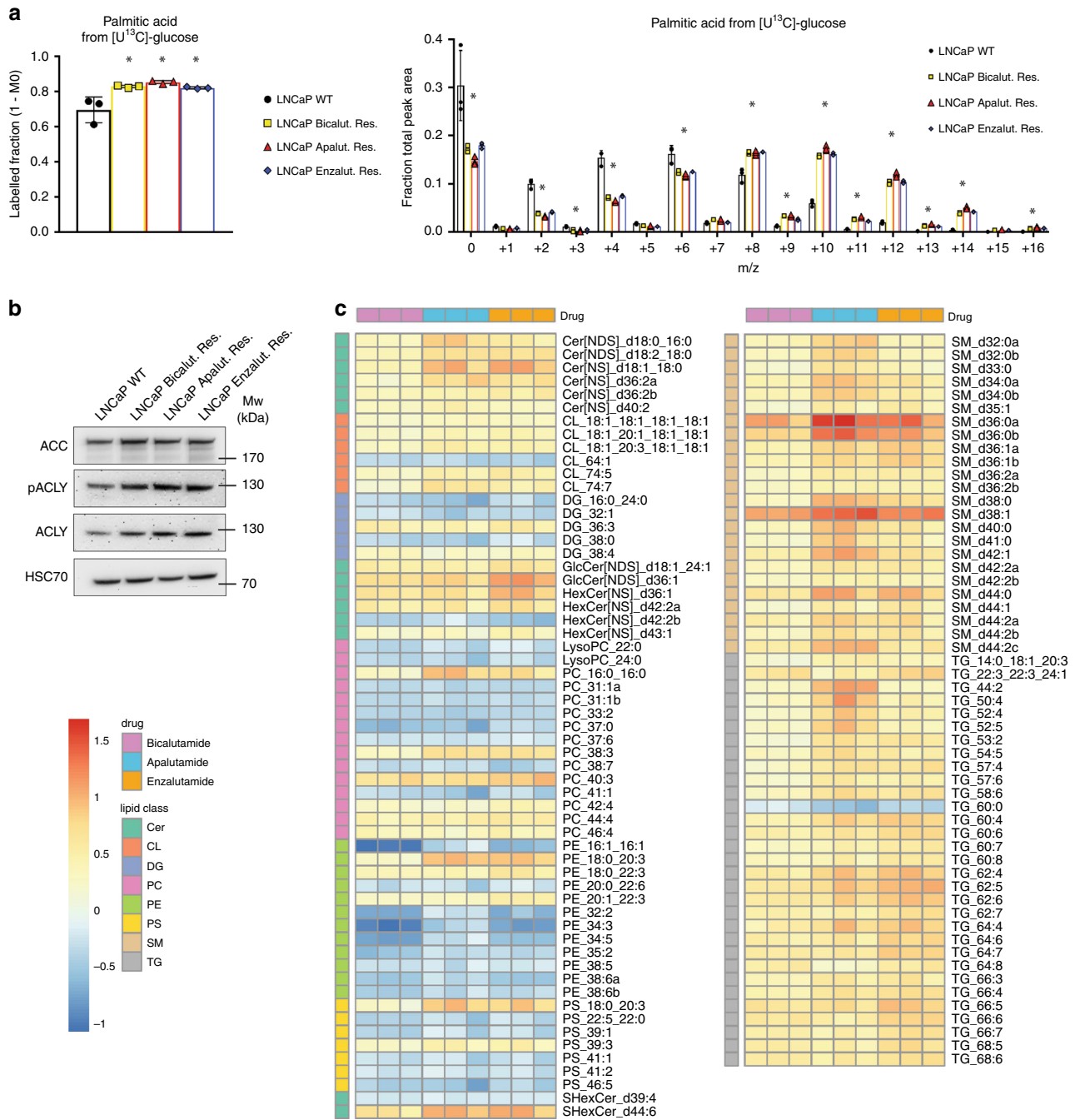

**Fig. 3 Lipid metabolism is strongly dysregulated in ARI-resistant cells. a** Labelled palmitate fraction derived from $^{13}$C-glucose (left panel) and relative isotopologue distribution of palmitic acid in ARI-resistant and WT LNCaP cells following $^{13}$C-glucose incubation for 72 h (right panel). **b** Western blot analysis of ACC, ACLY, and phospho-ACLY expression in WT and ARI-resistant LNCaP cells. HSC70 is used as a sample loading control. **c** Heatmap illustrating the steady-state levels of significantly regulated lipids in ARI-resistant cells when compared to WT LNCaP (FC > 1.5, $p < 0.05$ using a 1-way ANOVA with a Dunnett's multiple comparisons test). Values are expressed as log(FC). **a** Data are presented as mean values +/− SD. **a** *$p$-value < 0.05 using a 1-way ANOVA with a Dunnett's multiple comparisons test. Cer ceramide, CL cardiolipin, DG diacylglycerol, LysoPC lysophosphatidylcholine, PC phosphatidylcholine, PE phosphatidylethanolamine, PS phosphatidylserine, SM sphingomyelin, TG triglyceride. Source data are provided as a Source data File.

significantly enriched in comparison to negative control H19 (Supplementary Fig. 5c), suggesting that DECR1 might not be a direct AR-target.

Altogether, these results suggested that DECR1 expression might be a part of an adaptive response to AR inhibition and prompted us to further explore the role of this protein in the context of CRPC.

**DECR1 is associated with poor prognosis in CRPC.** In vivo, DECR1 expression was strongly upregulated in AR-proficient CRPC orthografts derived from LNCaP AI and VCaP CR cells grown under castrated condition (ADT) (Fig. 5f). Using immunohistochemistry, we further examined DECR1 protein expression on a tissue microarray (TMA) of patients that had been biopsied before and after they developed resistance to ADT. In

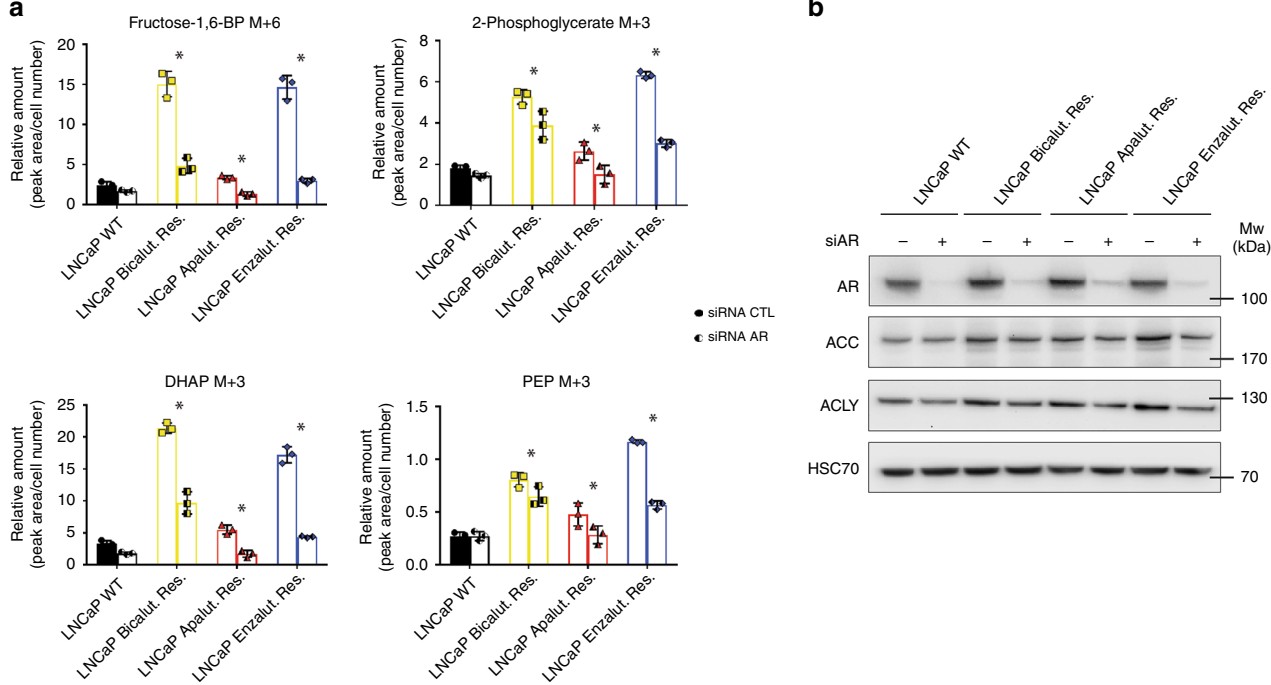

**Fig. 4 Increased glucose metabolism in ARI-resistant cells depends on AR signalling. a** Expression of labelled isotopologue of selected metabolites in ARI-resistant and WT LNCaP cells silenced for AR and following 1 h $^{13}$C-glucose incorporation. **b** Western blot analysis of AR, ACC and ACLY expression in WT and ARI-resistant LNCaP cells following AR siRNA silencing. HSC70 is used as a sample loading control. **a** Data are presented as mean values $+/-$ SD, *$p$-value of labelled fraction < 0.05 using a 2-way ANOVA with a Sidak's multiple comparisons test. DHAP dihydroxyacetone-phosphate, PEP phosphoenolpyruvate. Source data are provided as a Source data File.

the majority of the patients (9 out of 14), DECR1 staining was increased after recurrence (positive Δ histoscore) (Fig. 5g). Similarly, DECR1 expression was also upregulated in tumours persisting after castration therapy in the *Nkx3.1 Pten$^{fl+/-}$ Spry2$^{fl+/-}$* genetically engineered mouse model, an AR-driven model of CRPC[26] (Supplementary Fig. 5d).

We next assessed the clinical relevance of DECR1 in prostate cancer. In silico analysis of the cancer genome atlas (TCGA) dataset revealed that *DECR1* expression was frequently dysregulated in prostate cancer, at both gene and transcript levels (Fig. 5h). *DECR1* mRNA expression level was significantly higher in prostate tumour samples than in normal adjacent tissues, and high *DECR1* expression was associated with significantly reduced patient disease-free survival (Fig. 5i, j). In another dataset[27], high *DECR1* expression was observed in more than 70% of metastases collected from CRPC patients, independent of the site of metastasis (Supplementary Fig. 5e). Analysis of a third independent cohort[28] further confirmed that *DECR1* expression was highest in metastatic tumours (Supplementary Fig. 5f) and strongly correlated with worse outcome in metastatic patients (Supplementary Fig. 5g).

**DECR1 maintains lipid homeostasis in CRPC cells.** To evaluate the functional impact of DECR1 in CRPC, we used the isogenic pair of LNCaP (androgen-responsive) and LNCaP-AI (androgen-independent) cell lines. LNCaP AI cells displayed higher levels of both the DECR1 protein and mRNA than parental LNCaP (Fig. 6a, Supplementary Fig. 6a).

Using CRISPR-CAS9 gene editing technology, we deleted the *DECR1* gene in LNCaP-AI cells, referred to as DECR1 KO cells thereafter (Supplementary Fig. 6b). Loss of DECR1 expression impaired in vitro cellular proliferation of LNCaP-AI cells by an average of 32% (Fig. 6b). To evaluate the impact of DECR1

deletion on cell metabolism, we performed metabolomic profiling of small polar metabolites in DECR1 KO and CTL cells (Supplementary Fig. 6c). We did not observe any significant changes in the expression of TCA cycle metabolites or in the several carnitine derivatives that were identified (Supplementary Fig. 6d), which might indicate that loss of DECR1 does not strongly affect global β-oxidation or mitochondrial metabolism. Interestingly, levels of dihydroxyacetone-phosphate (DHAP) and glyceraldehyde-3-phosphate (G3P), two glycolytic intermediates, were lower in DECR1-deficient cells compared to CTL cells. In contrast, DECR1 KO accumulated larger amounts of glutamine (3.5-fold), which might be reflective of their lower proliferation rate, and phospho-ethanolamine (2-fold), an important precursor for the synthesis of phosphatidyl-ethanolamine (PE) and membrane lipids (Supplementary Fig. 6c). Intrigued by this result, we decided to compare the lipid profile of DECR1 KO cells with the one of CTL cells (Fig. 6c). In line with impaired PE biosynthesis, lipidomic analysis revealed decreased levels of several classes of PE in DECR1 KO cells. In addition, clustered analysis of the differentially altered lipids highlighted an imbalance between unsaturated and saturated lipids, with the DECR1 KO cells accumulating elevated levels of highly unsaturated lipids (saturation level >2) but low levels of saturated and mono-unsaturated or bi-unsaturated molecules (saturation level ≤ 2) (Fig. 6c). This result is exemplified by the dual distribution of altered phosphatidylcholines (PC) molecules in Supplementary Fig. 6e. DECR1-deficient cells also displayed high levels of ceramides (Fig. 6d), which have been shown to perturb cellular homeostasis and to be detrimental to mitochondrial function[29]. To better understand how DECR1 influences fatty acid (FA) metabolism and changes in FA saturation, we further performed GC-MS analyses to measure the FA composition of CTL and DECR1 KO cells. In comparison with CTL cells, DECR1

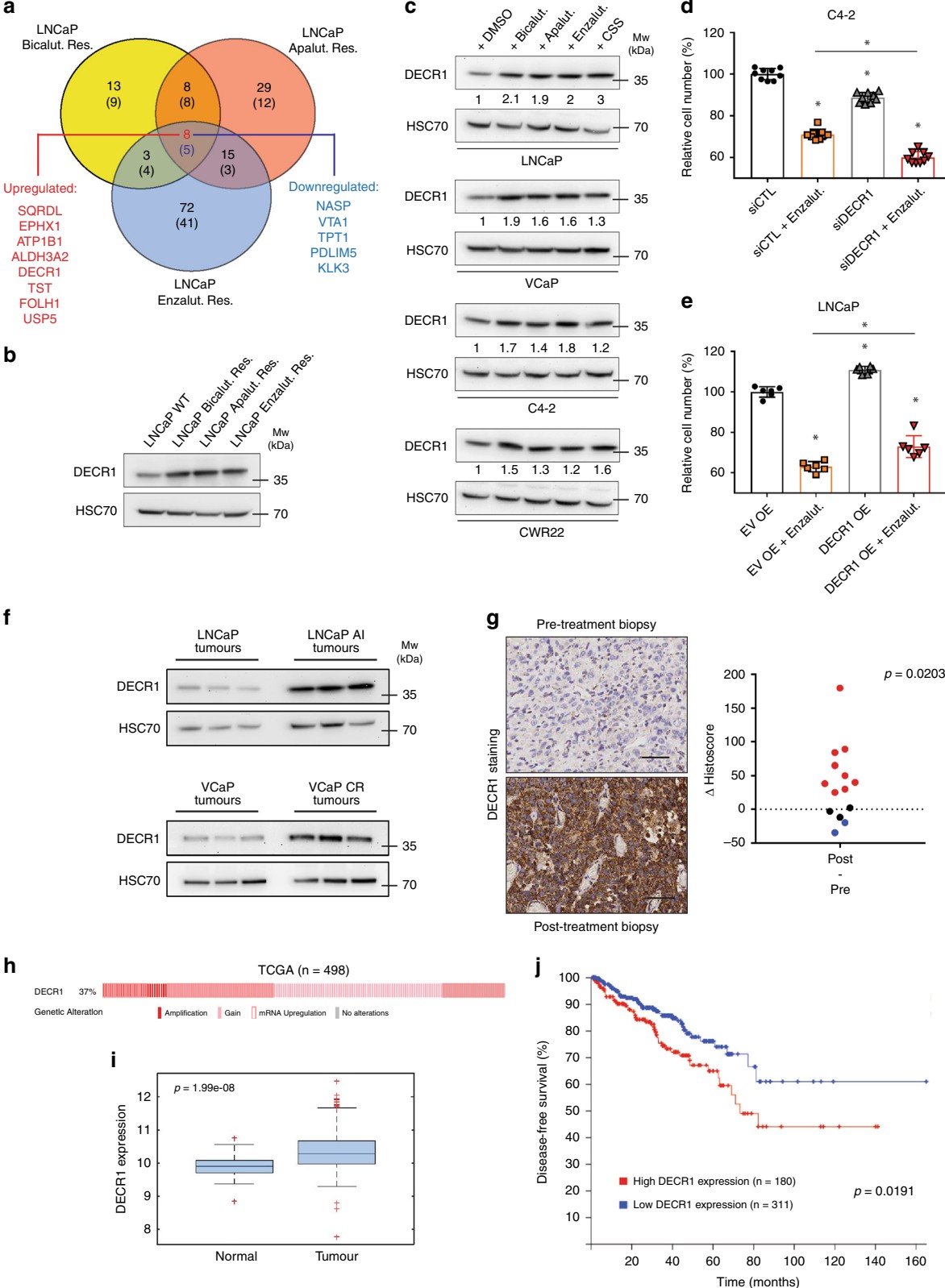

KO cells presented higher levels of polyunsaturated fatty acids (PUFAs), especially arachidonic acid (AA) and docosahexaenoic acid (DHA), while they showed decreased levels of the monounsaturated fatty acids (MUFAs) palmitoleic acid (PalA) and sapienate (SA) (Fig. 6e). While MUFAs have been shown to protect against oxidative stress[30], cellular accumulation of PUFAs can result in lipid peroxidation and induction of ferroptosis[31].

PUFA-induced toxicity mainly results from accumulation of free PUFAs. Therefore, we also compared the intracellular pool of free FA between the different conditions. Similar to the analysis of total FA, we found that DECR1 KO cells exhibited increased levels of free PUFAs (DHA and AA, although the latter only reached significance for one of the KO clones) and decreased levels of free MUFAs (PalA and SA) (Fig. 6f). This change in FA

**Fig. 5 DECR1 is a potential target for CRPC. a** Venn diagrams highlighting proteins commonly modulated ($p$-value < 0.05, FC > 1.5) in ARI-resistant cells cultured in 2D and 3D conditions. Upregulated proteins are on top; downregulated proteins are into brackets. **b** Western blot analysis of DECR1 expression in WT and ARI-resistant LNCaP cells. **c** Western blot analysis of DECR1 expression in $AR^+$ prostate cancer cells following acute AR inhibition for 48 h. **d** Cell proliferation of C4-2 cells silenced for DECR1 expression and treated with enzalutamide (20 μM—48 h). Cell count is normalised to untreated control (siCTL). **e** Cell proliferation of LNCaP cells overexpressing DECR1 and treated with enzalutamide (20 μM—48 h). Cell count is normalised to untreated empty vector (EV OE). **f** Western blot analysis of DECR1 expression in hormone naïve (LNCaP, VCaP) and castration-resistant (LNCaP AI, VCaP CR) tumour orthografts. **g** Immunohistochemical staining (left) and quantification (right) of DECR1 expression in CRPC tissue samples. Data are represented as difference in histoscore between post-treatment and pre-treatment biopsies. Scale bar represents 100 μm. **h** Percentage of prostate cancer patients showing genomic (copy number gain or amplification) or mRNA alteration (z-score = 1.5) for DECR1 using the TCGA dataset. **i** Gene expression analysis of DECR1 in normal and tumoural prostate tissues according to the TCGA dataset ($n = 498$). Centre line corresponds to median of data, top and bottom of box correspond to 75th and 25th percentile, respectively. Whiskers extend to adjacent values (minimum and maximum data points not considered outliers). **j** Kaplan–Meier survival analysis of prostate cancer patients stratified according to DECR1 expression using the TCGA dataset. **b**, **c**, **f** HSC70 is used as a sample loading control. **d**, **e** Data are presented as mean values +/− SD. **d**, **e** *$p$-value < 0.05 using a 1-way ANOVA with a Dunnett's multiple comparisons test. **g** statistical analysis was performed using a Wilcoxon matched-pair signed-rank test ($n = 14$). **i** statistical analysis was performed using a pairwise ANOVA. **j** statistical analysis was performed using a logrank test. Source data are provided as a Source data File.

| Gene name | Protein name | 2D Comparison (log$_2$(FC)) | | | 3D Comparison (log2(FC)) | | |
|---|---|---|---|---|---|---|---|
| | | Bicalut./WT | Apalut./WT | Enzalut./WT | Bicalut./WT | Apalut./WT | Enzalut./WT |
| SQRDL | Sulphide:quinone oxidoreductase, mitochondrial | 4.10 | 5.01 | 4.87 | 2.12 | 3.14 | 3.30 |
| EPHX1 | Epoxide hydrolase 1 | 1.31 | 1.81 | 2.55 | 0.78 | 2.29 | 2.62 |
| ATP1B1 | Sodium/potassium-transporting ATPase subunit beta-1 | 0.96 | 2.22 | 2.40 | 1.17 | 1.77 | 1.26 |
| ALDH3A2 | Fatty aldehyde dehydrogenase | 2.41 | 1.47 | 2.05 | 0.74 | 0.93 | 1.33 |
| DECR1 | 2,4-dienoyl-CoA reductase, mitochondrial | 1.08 | 1.73 | 1.28 | 1.23 | 1.76 | 1.09 |
| TST | Thiosulfate sulfurtransferase | 1.33 | 1.00 | 1.85 | 0.99 | 0.80 | 0.95 |
| FOLH1 | Glutamate carboxypeptidase 2 | 0.59 | 1.04 | 1.17 | 0.61 | 1.41 | 0.87 |
| USP5 | Ubiquitin carboxyl-terminal hydrolase 5 | 0.87 | 0.81 | 1.37 | 0.81 | 0.85 | 0.66 |
| NASP | Nuclear autoantigenic sperm protein | −1.05 | −1.41 | −1.34 | −0.72 | −0.99 | −0.75 |
| VTA1 | Vacuolar protein sorting-associated protein VTA1 homologue | −1.24 | −1.30 | −1.42 | −0.88 | −0.82 | −1.34 |
| TPT1 | Translationally-controlled tumour protein | −1.35 | −1.81 | −2.26 | −1.29 | −1.15 | −2.40 |
| PDLIM5 | PDZ and LIM domain protein 5 | −2.66 | −2.27 | −2.98 | −2.71 | −3.00 | −3.36 |
| KLK3 | Prostate-specific antigen | −2.63 | −2.83 | −2.45 | −4.39 | −3.79 | −3.61 |

**Table 1 Proteins commonly modulated in ARI (FC = 1.5, $p$ < 0.05) in 2D and 3D cultures.**

composition was not associated with differences in FA synthesis (Supplementary Fig. 6f). Finally, DECR1-deficient cells were less sensitive than CTL to the addition of palmitate, a saturated fatty acid (Supplementary Fig. 6h), and palmitate treatment was able to abolish DECR1-dependent decrease in cell proliferation (Supplementary Fig. 6i).

**Loss of DECR1 sensitises CRPC cells to ferroptosis.** Increased polyunsaturated lipid content can also increase susceptibility to lipid peroxidation and trigger ferroptosis. In line with this idea, loss of DECR1 resulted in increased expression of the ER chaperones BIP and DNAJC3, and lead to an activation of the unfolded protein response, a process known to be associated with impaired lipid homeostasis, as evidenced by strong upregulations of CHOP and the spliced isoform of XBP1 (Fig. 6g). DECR1 KO cells also displayed elevated levels of the lipid-detoxifying enzyme glutathione peroxidase 4 (GPX4)[32] (Fig. 6h). Therefore, we treated CTL and DECR1 KO cells with RSL3, a specific GPX4 inhibitor, to induce ferroptosis. DECR1 KO cells displayed increased sensitivity to GPX4 inhibition (Fig. 6i, Supplementary Fig. 6g). Co-treatment with either Trolox or liproxstatin, two inhibitors of ferroptosis, was able to rescue cell proliferation and restore cell morphology of DECR1 KO cells under RSL3 treatment (Fig. 6j, Supplementary Fig. 6g). Taken together, these results suggest a role for DECR1 in regulating redox homeostasis in a lipid-dependent manner.

**DECR1 expression is required for in vivo CRPC tumour growth.** In contrast to LNCaP, LNCaP-AI cells are able to form solid tumours when injected orthotopically into the prostate of castrated mice. Therefore, we examined the impact of reduced DECR1 expression on the growth of CRPC tumours in vivo. Silencing of DECR1 impaired tumour growth in castrated mice, significantly reducing tumour volume by an average of 41% (assessed by ultrasonography; Fig. 7a). Furthermore, DECR1 KO tumours were highly necrotic and displayed a lower level of cellularity than the control tumours (Fig. 7b).

Raman spectroscopy was performed on the tumour slides to test if loss of DECR1 expression perturbed intra-tumoural contents of lipids and cholesterol-based compounds. Raman spectroscopy is a non-destructive label-free analytical technique that can provide rich molecularly specific information without isolation of the specific molecules under investigation. Mixtures of molecules can be identified from their specific vibrations, with lipids in particular having a high Raman cross section making them ideal candidates for study by this technique. Lipid-associated molecules were represented by a band at 2845 cm$^{-1}$, while cholesterol content was demonstrated by a band around 2880 cm$^{-1}$[33]. Raman spectra obtained for DECR1 KO tumours demonstrated reduced intensity of bands at 2845 cm$^{-1}$ (lipids) and 2880 cm$^{-1}$ (cholesterol), suggesting that DECR1-deficient tumours had reduced levels of lipid and cholesterol compounds in comparison to control tumours (Fig. 7c). Finally, LC-MS

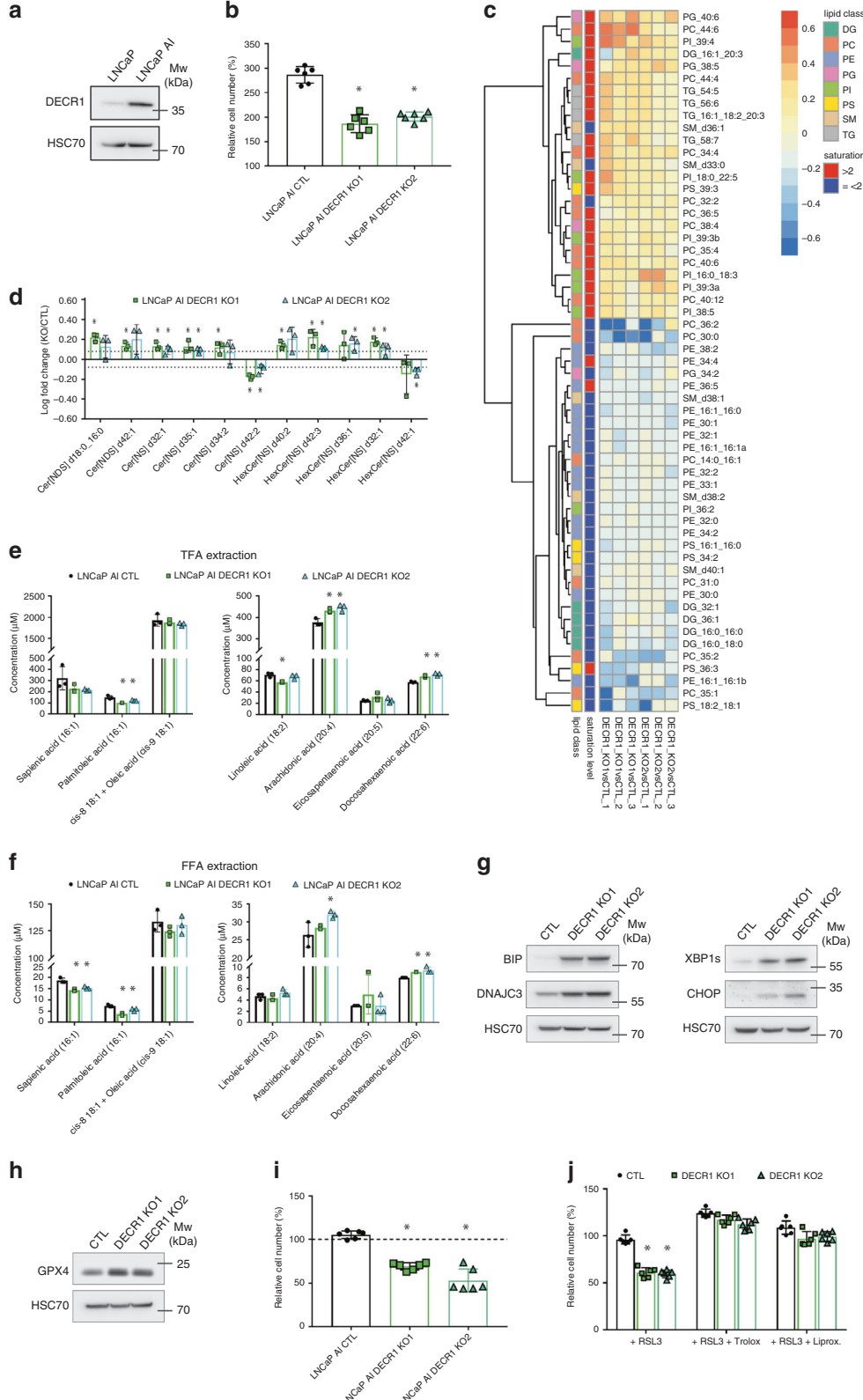

lipidomics performed on frozen tumours confirmed data from the Raman spectroscopy. DECR1 KO tumours presented a pronounced decrease in several highly abundant triglycerides species, which accounted for a large proportion of total lipid content (Fig. 7d). Moreover, similar to what is observed in vitro, DECR1 KO tumours showed an increase in several ceramides and multiple polyunsaturated phospholipids (Fig. 7d).

## Discussion

Despite being the main target of hormonal therapies for over five decades, AR remains the key driver of prostate cancer progression to CRPC. The recent clinical success of selective ARI in improving CRPC patient survival has further reinforced the importance of androgen receptor as a target for therapy. However, long-term resistance to AR-targeted therapies inevitably

**Fig. 6 DECR1 loss alters lipid homeostasis and sensitises CRPC cells to ferroptosis. a** Western blot analysis of DECR1 expression in LNCaP and LNCaP AI cells. **b** Cell proliferation of DECR1 KO (knockout) and untargeted control (CTL) cells after 72 h. Cell count is normalised to initial number of cells at the start of the experiment. **c** Hierarchical clustering of significantly altered lipids in DECR1 KO cells when compared to CTL cells (FC > 1.2). Selected lipids were significantly altered in at least one of the two KO cells ($p < 0.05$ using two-sided Student's $t$-test). Values are expressed as log(FC). **d** Steady-state levels of significantly altered ceramides (Cer) in DECR1 KO cells when compared to CTL cells (FC > 1.2). Selected lipids were significantly altered in at least one of the two KO cells ($p < 0.05$ using two-sided Student's $t$-test). **e** Absolute concentration of total MUFAs (left panel) and total PUFAs (right panel) in CTL and DECR1 KO cells, quantified by GC-MS. **f** Absolute concentration of free MUFAs (left panel) and free PUFAs (right panel) in CTL and DECR1 KO cells, quantified by GC-MS. **g** Western blot analysis of BIP, DNAJC3, XBP1s and CHOP expression in DECR1 KO and CTL cells. **h** Western blot analysis of GPX4 expression in DECR1 KO and CTL cells. **i** Cell proliferation of DECR1 KO and CTL cells treated for 48 hours with RSL3 (10 µM). Cell count is normalised to initial number of cells. **j** Cell proliferation of DECR1 KO and CTL cells treated for 48 h with RSL3 (10 µM) and Trolox (20 µM) or Liproxstatin (50 nM). Cell count is normalised to initial number of cells. **a, g, h** HSC70 is used as a sample loading control. **b, d, e, f, i, j** Data are presented as mean values +/− SD. **b, e, f, i, j** *$p$-value < 0.05 using a 1-way ANOVA with a Dunnett's multiple comparisons test. DG: diacylglycerol, PC phosphatidylcholine, PE phosphatidylethanolamine, PG phosphatidylglycerol, PI phosphatidylinositol, PS phosphatidylserine, SM sphingomyelin, TG triglyceride. Source data are provided as a Source data File.

leads to tumour relapse, and therefore a better understanding of the molecular mechanisms underlying therapy resistance represents an urgent clinical need and a top research priority.

Cancer cells have developed multiple ways to escape AR-directed targeting. While situations of AR loss[34] or bypass[17] have been reported, strong evidences suggest that AR signalling remains active in the majority of CRPC patients[35]. Along with the acquisition of point mutations[12–14], *AR* gene amplification as well as overexpression at mRNA and protein level[36] has been suggested to support enzalutamide resistance by activating AR-dependent and independent pathways[37,38]. In line with these reports, we found that AR signalling remained highly active in cells resistant to bicalutamide, apalutamide and enzalutamide. AR transcriptional reprogramming was evidenced by AR nuclear localisation, and varying levels of expression for canonical AR targets (namely KLK3 and FKBP5).

ARI-resistant cells had their metabolism altered towards increased glucose consumption. In support of our findings, increased glucose metabolism has been associated with late stage CRPC[20], and inhibition of glycolysis synergised with enzalutamide treatment[39]. We initially thought that a potential driver for this metabolic rearrangement could be the master metabolic sensor AMPK, which has been proposed as a target in the context of enzalutamide resistance[24,40]. However, while ARI-resistant cells indeed showed increased AMPK expression, silencing of this protein only moderately affected glucose utilisation. In contrast, AR silencing dramatically reduced the glucose flux in the resistant cells, suggesting that this metabolic reprogramming depends on AR itself. AR is an important regulator of cell metabolism in the prostate[18]. In an elegant study, Massie and colleagues demonstrated that AR could directly regulate aerobic glycolysis and glucose-derived anabolism under conditions of androgen stimulation by activating the transcription of key biosynthetic enzymes such as HK1, HK2, ACC, and FASN. Interestingly, tracing of $^{13}$C-glucose incorporation into fatty acids revealed that the higher rate of glucose oxidation observed in ARI-resistant cells contributes to enhanced de novo fatty acid synthesis. In addition to reducing the glucose flux, AR silencing in these cells also led to a selective downregulation of the lipogenic enzymes ACLY and ACC. Thus, the metabolic phenotype observed in ARI-resistant LNCaP cells is highly reminiscent of the one observed in DHT-activated LNCaP[18], and therefore represents a metabolic vulnerability of therapeutic potential. In line with this idea, pharmacological inhibition of the mitochondrial pyruvate carrier has recently been demonstrated to specifically impair AR-driven tumour growth[41].

In addition to glucose metabolism, our proteomic analysis revealed that many mitochondrial proteins were overexpressed in the resistant cells. The metabolic rewiring observed in ARI-resistant cells ultimately led to an important reorganisation of the lipidome of these cells, characterised by a strong accumulation of polyunsaturated triglycerides and sphingolipids. Therefore, while the influence of mitochondria on ARI resistance remains to be fully tested, we can hypothesise that increased mitochondrial metabolism in the resistant cells was needed to support FA and lipid synthesis, via the generation of citrate and acetyl-CoA, and also to regulate lipid homeostasis, for example through an increase in β-oxidation. This hypothesis is further supported by the high expression of several transcription factors involved in mitochondrial metabolism and lipid homeostasis, such as PGC1α, PPARγ and LXR, in ARI-resistant cells. Both lipid synthesis and fatty acid degradation are highly dysregulated processes in prostate cancer[20], and these pathways have been the subject of intense drug development[42,43]. On one hand, targeting enzymes involved in de novo lipid synthesis, such as ACLY[44] or FASN, is an active research focus with Phase 2 clinical trials currently opened[42]. On the other hand, fatty acid oxidation (FAO) can promote cancer cell survival under stress conditions[45,46] and pharmacological inhibition of that pathway proved effective in multiple cancers[47,48]. In prostate, increased beta-oxidation has been shown to promote prostate cancer cell survival[49] while inhibition of lipid catabolism, achieved via pharmacological inhibition or CPT1a silencing, decreased tumour growth and restored enzalutamide sensitivity[50,51].

Lastly, our proteomic analysis highlighted DECR1, an auxiliary enzyme of β-oxidation, as a protein robustly associated with ARI resistance. In addition, ARI-resistant cell lines were characterised by high levels of multiple metabolic enzymes involved in lipid metabolism (DECR2, ACSL1, HIBCH, HIBADH, etc.). Therefore, we speculated that DECR1 was overexpressed upon resistance to AR inhibition as part of an adaptive response that allows resistant cells to cope with their rewired lipid metabolism. High DECR1 expression has previously been detected in proteomic analyses comparing localised prostate tumours to non-malignant lesions[52] or in metastatic specimens compared to localised tumours[53]. In addition to that, we demonstrated that high expression of DECR1 in primary and metastatic prostate cancer correlates with unfavourable patient survival outcomes. Short term ARI treatment or androgen withdrawal induced DECR1 expression in prostate cell lines, while long-term ADT treatment in patients and tumour-bearing mice resulted in strong upregulation of the protein. However, DECR1 levels in prostate cells were not affected by AR silencing or DHT treatment, suggesting that DECR1 might be part of a long-term, rather than acute, response to AR inhibition. Future studies will be needed to understand the mechanisms involved in the regulation of DECR1 during CRPC development.

DECR1 is a mitochondrial enzyme involved in the auxiliary pathway of beta-oxidation and critically regulates PUFAs in entering the degradation cycle within the mitochondria. To date,

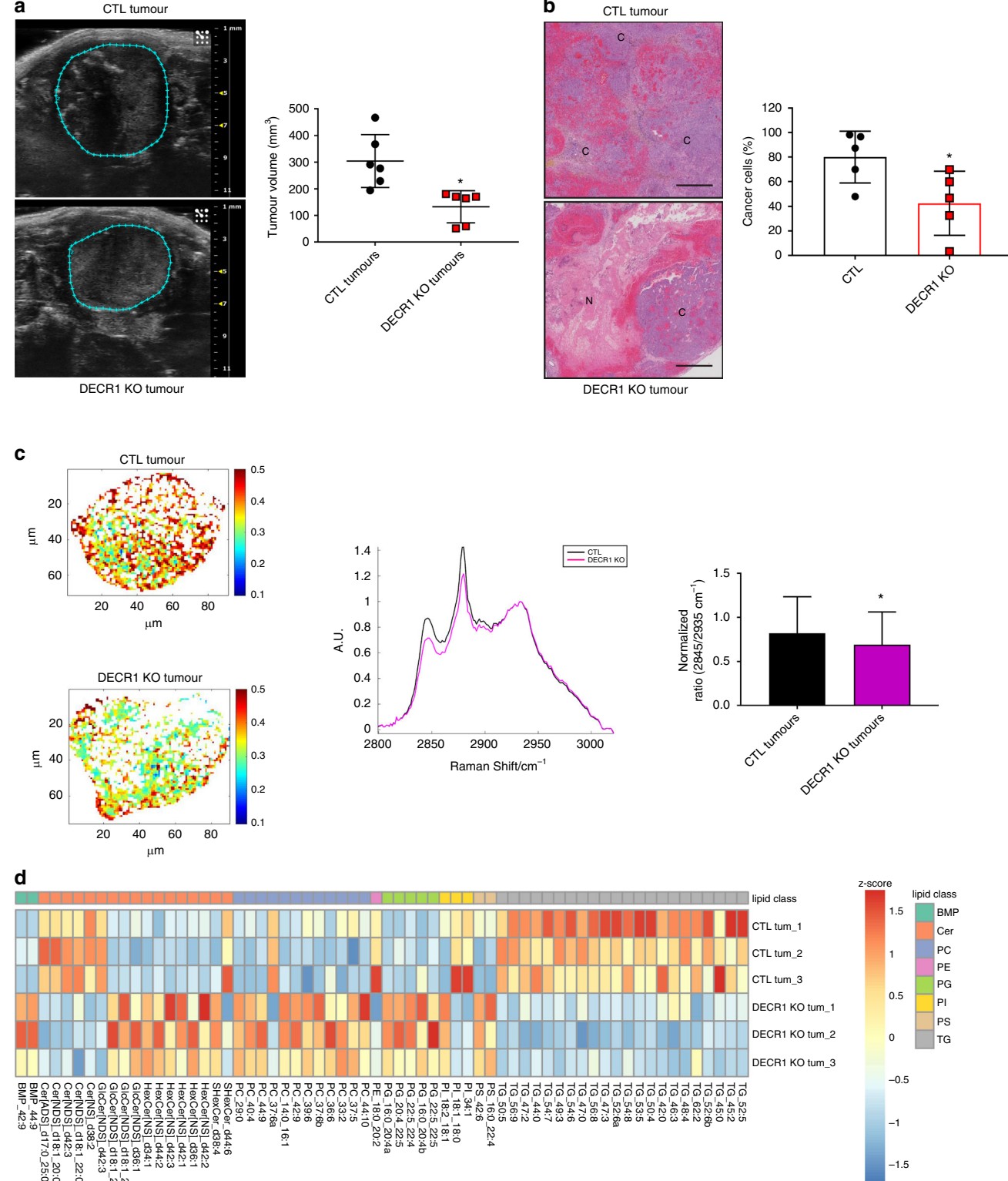

the role of DECR1 in cancer has not been well-studied. In contrast to our data, DECR1 was proposed to act as a tumour suppressor in HER2-positive breast cancer[54]. However, in the same study, DECR1 overexpression was also shown to protect cancer cells from glucose withdrawal-induced apoptosis. Interestingly, *Decr1*[−/−] mice develop hypoglycaemia under metabolic stress conditions[55], suggesting a link between DECR1 and glucose metabolism. We also observed decreased levels of the glycolytic

intermediates DHAP and G3P in DECR1-deficient cells (Supplementary Fig. 6c). However, the main phenotype described in *Decr1*[−/−] mice was the disturbance of fatty acid balance, with *Decr1*[−/−] mice showing accumulation of unsaturated fatty acids in the form of liver triacylglycerols[55]. Therefore, in the context of CRPC, an appealing hypothesis would be that DECR1-mediated lipid rearrangement is necessary to support the rewired metabolism of ARI-resistant cells, and its associated stress stimuli

**Fig. 7 DECR1 KO affects lipid metabolism and decreases CRPC tumour growth. a** Representative pictures of LNCaP AI CTL (top image) or DECR1 KO (bottom image) tumour orthografts monitored by ultrasound imaging (left). Quantification of tumour volume using ultrasonography (right). **b** Representative pictures of hematoxylin/eosin staining on orthografts from LNCaP AI CTL (top) or DECR1 KO (bottom) cells. C = cancer cells, N = necrotic region. Quantification of the proportion of cancer cells (right). Scale bar represents 1000 μm. **c** Representative heatmaps on data from Raman spectroscopy using 532 nm excitation (intensity at frequency 2845 cm$^{-1}$/intensity at frequency 2935 cm$^{-1}$) on LNCaP AI CTL (top) and DECR1 KO (bottom) derived orthografts (left). Average Raman spectra of LNCaP AI CTL (grey) or DECR1 KO (magenta) orthografts (middle) and quantification of tumour lipid content using the 2845 cm$^{-1}$-peak (right panel). AU = arbitrary unit. **d** Heatmap illustrating the steady-state levels of significantly regulated lipids in DECR1-deficient tumours when compared to CTL tumours (FC > 1.5, $p < 0.05$ using a two-tailed Student's $t$-test). Values are expressed as z-score. **a–c** Data are presented as mean values +/− SD. **a–c** *p-value < 0.05 using a two-tailed Mann–Whitney U-test. BMP Bis(monoacylglycero)phosphate, Cer ceramide, PC phosphatidylcholine, PE phosphatidylethanolamine, PG phosphatidylglycerol, PI phosphatidylinositol, PS phosphatidylserine, TG triglyceride. Source data are provided as a Source data File.

(e.g., redox stress). Similarly to what is observed in *Decr1*$^{-/-}$ mice, we were able to demonstrate that DECR1 was critically required for lipid homeostasis in CRPC. Indeed, DECR1 KO cells accumulated high levels of polyunsaturated lipids and low levels of saturated lipids compared to control cells. This observation was further supported by an accumulation of total and free PUFAs together with a decrease in several MUFAs, such as the recently-reported MUFA sapienate[56], in DECR1-deficient cells. Interestingly, we did not observe any change in the levels of the highly abundant MUFA oleic acid (18:1). Because DECR1 acts primarily on PUFAs; this result suggests that changes in PUFA composition, rather than MUFA, might contribute to a greater extent to the generation of the DECR1 KO phenotype. This imbalance in turn resulted in an activation of the ER stress response pathway and an increased susceptibility to lipid peroxidation, evidenced by the observed increased sensitivity to GPX4 inhibition. Accumulation of PUFAs increases the risk of lipid peroxidation and cell death[31], and pharmacological inhibition of GPX4, an important lipid-detoxifying enzyme, has recently been proposed as a promising therapeutic strategy in drug-resistant tumours[32]. Moreover, specific PUFA-derived mediators, such as specialised pro-resolving mediators, have recently attracted interest due to their anti-tumoural properties[57]. Therefore, by regulating lipid saturation level, DECR1 might act in several ways to support cancer cell survival. Finally, changes in membrane-lipid saturation are known to cause ER stress[58,59] and, interestingly, DECR1 has been reported to be a strong interacting partner of DNaK, the homologue of the Hsp70 chaperone in bacteria[60]. Taken together, these results suggest that DECR1 may contribute to increase metabolic flexibility of cancer cells under stress conditions, and uncover an unexpected link between DECR1 function and ER homeostasis.

In conclusion, our data suggest that, despite differences in available AR-targeted therapies, castration-resistant prostate cancer cells develop a shared metabolic phenotype in response to AR inhibition. In addition to being directly targetable, these metabolic adaptations are associated with the emergence of a specific set of proteins, such as DECR1, which could serve as clinically relevant biomarkers and/or therapeutic targets. These findings support the idea of targeting specific metabolic pathways in combination with AR inhibition, and open doors for the development of future therapies.

## Methods
**2D and 3D cell culture**. All cells were cultured in RPMI (Gibco, Thermo Fisher Scientific, Waltham, MA, USA) medium supplemented with 10% foetal bovine serum (FBS, Gibco, Thermo Fisher Scientific, Waltham, MA, USA) and 2 mM glutamine (Gibco, Thermo Fisher Scientific, Waltham, MA, USA), and maintained at 37 °C under 5% CO$_2$. LNCaP (ATCC CRL-1740) and C4-2 cells (ATCC CRL3314) were obtained from ATCC. CWR22Res cells (hormone-responsive variant of CWR22 cells) were obtained from Case Western Reserve University, Cleveland, Ohio. The bicalutamide-resistant and enzalutamide-resistant LNCaP derivatives (LNCaP Bicalut. Res. and LNCaP Enzalut. Res.) were generated by

chronic treatment of LNCaP cells with respective 10 μM bicalutamide (Sigma Aldrich, St Louis, MI, USA) or 10 μM enzalutamide (Selleckchem, Munich, Germany) for 4 months until non-clonal surviving populations of cells were propagated. For the apalutamide-resistant LNCaP cell line (LNCaP Apalut. Res.), cells were grown as above in 10 μM apalutamide (ARN509, Selleckchem, Munich, Germany) and a clonal population of surviving cells were expanded. All LNCaP derivatives were grown routinely in 10 μM of their respective anti-androgen.

LNCaP-AI cells were maintained in phenol-free RPMI (Gibco, Thermo Fisher Scientific, Waltham, MA, USA) supplemented with 10% charcoal-stripped serum (Gibco, Thermo Fisher Scientific, Waltham, MA, USA) and 2 mM glutamine.

For 3D culture, cells were suspended in 100% Matrigel (Corning, NY, USA) and seeded in 6-wells plate. Matrigel was allowed to solidify for 15 min at 37 °C before being entirely covered with 3 ml of culture medium. Cells were allowed to grow for five passages before each experiment. For each passage, Matrigel was mechanically disrupted and organoids were gently centrifuged at 300×g for 5 min. The organoid pellet was washed twice with PBS, and ¼ was finally resuspended in 100% Matrigel.

**Proteomic analysis**. 6–8 × 10$^5$ cells were seeded in 6-well plates and allowed to attach overnight. The next day, culture medium was replaced with fresh RPMI and cells were allowed to grow for another 48 h. Cells were then washed with PBS and lysed in 8 M Urea. Reduced proteins were alkylated using 55 mM Iodoacetamide for 1 h at room temperature. Alkylated proteins were then submitted to a two-step digestion at 35 °C with endoproteinase Lys-C (Alpha Laboratories, Eastleigh, UK) for 1 h, followed by trypsin (Promega, Madison, WI, USA) digestion overnight.

Digested peptides were desalted using StageTip[61] and separated by nanoscale C18 reverse-phase liquid chromatography performed on an EASY-nLC II (Thermo Fisher Scientific, Waltham, MA, USA) coupled to an Orbitrap Velos mass spectrometer (Thermo Fisher Scientific, Waltham, MA, USA). Elution was carried out using a binary gradient with water (buffer A) and 80% acetonitrile (buffer B), both containing 0.1% of formic acid. Peptide mixtures were separated at 200 nl min$^{-1}$ flow, using a 20 cm fused silica emitter (New Objective) packed in-house with ReproSil-Pur C$_{18}$-AQ, 1.9 μm resin (Dr. Maisch GmbH). Packed emitter was kept at 35 °C by means of a column oven integrated into the nanoelectrospray ion source (Sonation, Biberach, Germany). The gradient used started at 2% of buffer B, kept at same percentage for 5 min, then increased to 30% over 90 min and then to 60% over 15 min. Finally, a column wash was performed ramping to 80% of B in 5 min and then to 95% of B in 1 min followed by a 13 min re-equilibration at 2% B for a total duration of 129 min. The eluting peptide solutions were automatically (online) electrosprayed into the mass spectrometer via a nanoelectrospray ion source (Thermo Fisher Scientific, Waltham, MA, USA). An Active Background Ion Reduction Device (ABIRD, ESI Source Solutions, Woburn, MA, USA) was used to decrease ambient contaminant signal level.

Samples were acquired on a Linear Trap Quadrupole—Orbitrap Velos Mass spectrometer using a spray voltage, 2.4 kV, and an ion transfer tube temperature of 200 °C. The mass spectrometer was operated in positive ion mode and used in data-dependent acquisition mode (DDA). A full scan (FT-MS) was acquired at a target value of 1,000,000 ions with resolution $R = 60,000$ over mass range of 350–1600 amu. The top ten most intense ions were selected for fragmentation in the linear ion trap using collision induced dissociation (CID) using a maximum injection time of 25 ms or a target value of 5000 ions. Multiply charged ions from two to five charges having intensity greater than 5000 counts were selected through a 2 amu window and fragmented using normalised collision energy of 36 for 10 ms. Former target ions selected for MS/MS were dynamically excluded for 25 s.

The MS Raw files were processed with MaxQuant software[62] version 1.5.5.1 and searched with Andromeda search engine[63], querying UniProt[64] *Homo sapiens* (09/07/2016; 92,939 entries). The database was searched requiring specificity for trypsin cleavage and allowing maximum two missed cleavages. Methionine oxidation and N-terminal acetylation were specified as variable modifications, and cysteine carbamidomethylation as fixed modification. The peptide, protein and site false discovery rate (FDR) was set to 1%. Proteins were quantified according to the label-free quantification algorithm available in MaxQuant[65]. MaxQuant output was further processed and analysed using Perseus software version 1.5.5.3[66]. The common reverse and contaminant hits (as defined in MaxQuant output) were

removed. Only protein groups identified with at least one uniquely assigned peptide were used for the analysis. Significantly enriched proteins were selected using a two-sided $t$-test analysis with a 5% FDR.

**Metabolomic analysis**. $6-8 \times 10^5$ cells were seeded in 6-well plates and allowed to attach overnight. The next day, culture medium was replaced with fresh RPMI medium, and cells were allowed to grow for another 48 h. For tracing experiments, cells were further incubated in glucose-free RPMI (Gibco, Thermo Fisher Scientific, Waltham, MA, USA) supplemented with 10% FBS and 10 mM $^{13}$C-glucose (CLM-1396, Cambridge Isotopes Laboratories, UK) prior to extraction. Cells were then washed twice with ice-cold PBS and polar metabolites were extracted in a MetOH-Acetonitrile-H$_2$O (50-30-20) buffer for 5 min under gentle agitation. Collected supernatant was vigorously mixed for 10 min and centrifuged at 16,100×$g$ for ten additional minutes. All the steps were performed at 4 °C. Finally, supernatant was transferred to glass HPLC vials (Thermo Fisher Scientific, Waltham, MA, USA) and analysed using HPLC-MS.

Samples were analysed according to the refs. [67,68]. A Q Exactive Orbitrap mass spectrometer (Thermo Fisher Scientific, Waltham, MA, USA) was used together with a Thermo Ultimate 3000 HPLC system. The HPLC setup consisted of a ZIC-pHILIC column (SeQuant, 150 × 2.1 mm, 5 μm, Merck KGaA, Darmstadt, Germany), with a ZIC-pHILIC guard column (SeQuant, 20 × 2.1 mm) and an initial mobile phase of 20% 20 mM ammonium carbonate, pH 9.2, and 80% acetonitrile. Cell and media extracts (5 μl) were injected and metabolites were separated over a 15 min mobile phase gradient, decreasing the acetonitrile content to 20%, at a flow rate of 200 μl min$^{-1}$ and a column temperature of 45 °C. The total analysis time was 25 min. All metabolites were detected across a mass range of 75–1000 $m/z$ using the Q Exactive mass spectrometer at a resolution of 35,000 (at 200 $m/z$), with electrospray ionisation (ESI) and polarity switching to enable both positive and negative ions to be determined in the same run. Lock masses were used and the mass accuracy obtained for all metabolites was below 5 ppm. Data were acquired with Thermo Xcalibur software.

The peak areas of different metabolites were determined using Thermo TraceFinder 4.0 software, where metabolites were identified by the exact mass of the singly charged ion and by known retention time on the HPLC column. Commercial standards of all metabolites detected had been analysed previously on this LC-MS system with the pHILIC column. The $^{13}$C labelling patterns were determined by measuring peak areas for the accurate mass of each isotopologue of many metabolites. Intracellular metabolites were normalised to protein content of the cells, measured at the end of the experiment by the BCA assay (Thermo Fisher Scientific, Waltham, MA, USA).

**Lipidomic analysis**. Lipids were extracted from cells with 0.8 ml 1:1 butanol–methanol after media removal. Before extraction, an internal standard (Splash Lipidomix, Avanti Polar Lipids, Alabaster, AL, USA) was added to the extraction buffer and further used as a quality control. Lipids were separated on an Acquity UPLC CSH C18 column (100 × 2.1 mm; 1.7 μm) (Waters corporation, Milford, MA, USA) maintained at 60 °C. The mobile phases consisted of 60:40 ACN:H$_2$O with 10 mM ammonium formate, 0.1% formic acid and 5 μM of phosphoric acid (A) and 90:10 IPA:ACN with 10 mM ammonium formate, 0.1% formic acid and 5 μM phosphoric acid (B). The gradient was as follows: 0–2 min 30% (B); 2–8 min 50% (B); 8–15 min 99% (B), 15–16 min 99% (B), 16–17 min 30% (B). Sample temperature was maintained at 6 °C in the autosampler and 2 μl of sample were injected.

Analysis of polar and non-polar lipids were conducted using an LC-MS system including an Ultimate 3000 HPLC (Thermo Fisher Scientific, Waltham, MA, USA) coupled to a Q-Exactive Orbitrap mass spectrometer (Thermo Fisher Scientific, Waltham, MA, USA). Q-Exactive Orbitrap MS instrument was operated in both positive and negative polarities, using the following parameters: mass range 240–1200 $m/z$ (positive) and 240–1600 (negative), spray voltage 3.8 kV (ESI+) and 3 kV (ESI−), sheath gas (nitrogen) flow rate 60 units, auxiliary gas (nitrogen) flow rate 25 units, capillary temperature (320 °C), full scan MS1 mass resolving power 70,000. Data dependent fragmentation (dd-MS/MS) parameters for each polarity as follows: TopN: 10, resolution 17,500 units, maximum injection time: 25 ms, automatic gain control target: 5e$^5$ and normalised collision energy of 20 and 25 (arbitrary units) in positive polarity. TopN: 5, resolution 17,500 units, maximum injection time: 80 ms automatic gain control target: 5e$^5$ and normalised collision energy of 20 and 30 (arbitrary units) in negative polarity. The instrument was externally calibrated to <1 ppm using ESI positive and negative calibration solutions (Thermo Fisher Scientific, Waltham, MA, USA). Peak detection and integration from Raw data were processed using Compound Discoverer 3.0 (Thermo Fisher Scientific, Waltham, MA, USA). Files were also converted to mgf format using MSConvert software and MS2 files were searched against LipidBlast database using LipiDex software[69–71]. Peak intensities were normalised to cell number. Heatmaps were generated in R (v.3.6.1, https://www.R-project.org) using pHeatmap package (v.1.0.12, https://CRAN.R-project.org/package=pheatmap).

**Total and free fatty acid extraction and GC-MS**. Metabolite quenching and extraction were performed according to the ref. [72–74]. Every step was performed in a mixture of dry and wet ice. Briefly, cell plates previously quenched in liquid

nitrogen were extracted with 800 μl of cold 62.5% methanol. For the measurement of free fatty acids, two wells per sample were pooled. Cells were scraped with a pipet tip and transferred to Eppendorf tubes. Thus, 500 μl of cold chloroform containing 10 μg ml$^{-1}$ of C17 internal standard was added and samples were vortexed at 4 °C for 10 min. Phase separation was achieved by centrifugation at 4 °C for 10 min, after which the chloroform phase (lower phase containing fatty acids), the polar metabolite phase and the protein layout were separated and dried overnight with a vacuum concentrator at 4 °C. Standard curves at a final concentration of 2 μM to 800 μM were also extracted for the quantification of free and total fatty acids.

Total fatty acid samples and standards were esterified with 500 μl 2% sulfuric acid in methanol overnight at 50 °C, while free fatty acid samples and standards were esterified for 15 min at room temperature. Free and total fatty acids were then extracted by addition of 600 μl hexane and 100 μl saturated NaCl. Samples were centrifuged for 5 min and the hexane phase was separated and dried with a vacuum concentrator Samples and standards were resuspended in 50 μl of hexane.

Fatty acids were separated with gas chromatography (8860 GC system, Agilent Technologies, CA, USA) combined with mass spectrometry (5977B Inert MS system, Agilent Technologies, CA, USA). One microliter of each sample was injected (splitless mode) with an inlet temperature of 270 °C onto a DB-FASTFAME column (30 m × 0.250 mm). Helium was used as a carrier gas with a flow rate of 1 ml min$^{-1}$. For the separation of fatty acids, the initial gradient temperature was set at 50 °C for 1 min and increased at the ramping rate of 12 °C min$^{-1}$ to 180 °C, following by a ramping rate of 1 °C min$^{-1}$ to reach 200 °C. Finally, the final gradient temperature was set at 230 °C with a ramping rate of 5 °C min$^{-1}$ for 2 min. The temperatures of the quadrupole and the source were set at 150 °C and 230 °C, respectively. The MS system was operated under electron impact ionisation at 70 eV and a mass range of 100–600 amu was scanned. After the acquisition by GC-MS, an inhouse Matlab M-file was used to extract mass distribution vectors, to integrate raw ion chromatograms and to correct by isotopologue distributions by the natural abundance[75]. For the quantification of free and total fatty acids, the total ion counts were normalised to the internal standard (C17) and protein contents for cell extracts.

**Determination of $^{13}$C fatty acids and GC-MS**. $6-8 \times 10^5$ were seeded in 6-well plates and allowed to attach overnight. The next day, culture medium was replaced and cells were incubated in glucose-free RPMI (Gibco, Thermo Fisher Scientific, Waltham, MA, USA) supplemented with 10% FBS, 2 mM glutamine and 10 mM $^{13}$C-glucose (CLM-1396, Cambridge Isotopes Laboratories, UK) for another 72 h prior to extraction. Cells were then washed three times with ice-cold PBS and fatty acids were extracted in a Methanol-Chloroform-PBS buffer (750 μl 1:1 v/v PBS:methanol and 500 μl chloroform). For all samples, 50 μl of 1 mg ml$^{-1}$ methanolic butylated hydroxytoluene (BHT, Sigma Aldrich, St Louis, MI, USA) and 20 μl of 0.05 mg ml$^{-1}$ 17:0 PC (Avanti Polar Lipids, Albaster, AL, USA) as an internal standard, were added. Samples were centrifuged at 10,000×$g$ for 5 min before the lower chloroform layer was extracted and dried under N$_2$. Samples were reconstituted in 90 μl chloroform and incubated with 10 μl MethPrepII (Thermo Fisher Scientific, Waltham, MA, USA) for 20 min at room temperature.

Fatty acid methyl esters (FAMEs) were analysed using an Agilent 7890B GC system coupled to a 7000 Triple Quadrupole GC-MS system, with a Phenomenex ZB-1701 column (30 mm × 0.25 mm × 0.25 μm). An initial temperature of 45 °C was set to increase at 9 °C min$^{-1}$, held for 5 min, then 240 °C min$^{-1}$, held for 11.5 min, before reaching a final temperature of 280 °C min$^{-1}$, held for 2 min. The instrument was operated in pulsed splitless mode in the electron impact mode, 50 eV, and mass ions were integrated for quantification using known standards to generate a standard curve. Palmitic, stearic and oleic acid peak areas were extracted using mass-to-charge ratios ($m/z$) 270, 298 and 296, respectively. Mass Hunter B.06.00 software (Agilent) was used to quantify isotopomer peak areas before natural abundance isotope correction was performed using an in-house algorithm.

**Bioinformatics analysis**. Gene expression data were downloaded from TCGA and the GEO website. TCGA RNASeqV2 data were shift log transformed and GSE21034 data were log transformed, using mean of probes per gene. Expression values were grouped according to sample type and group distributions were plotted using the matlab routine boxplot with the bar indicating the median, the box spanning from the 25th to the 75th percentiles, and whiskers spanning 2.7\sigma. Outliers beyond that span are indicated in red. Significant difference between the groups was measured using overall ANOVA.

For survival analysis, gene expression data from human samples (excluding cell lines) was normalised as before, mean-centred, and clustered into three groups using k-means. Kaplan–Meier survival curves for the groups were plotted using the matlab routine kmplot.

**Human prostate cancer orthografts**. In vivo orthograft experiments were performed in accordance with the ARRIVE guidelines[76], and by a local ethics committee under the Project Licence P5EE22AEE in full compliance with the UK Home Office regulations (UK Animals (Scientific Procedures) Act 1986). Briefly, $20 \times 10^6$ cells/mouse were suspended in serum-free medium and mixed with Matrigel (Corning, NY, USA) in a 1:1 ratio. 50 μl of cell suspension were injected

orthotopically into the anterior prostate lobe of CD1-nude mice (Charles River Laboratories, Wilmington, MA, USA). Orchidectomy was performed at the time of injection. Tumours were allowed to grow for ~6 weeks after injection and tumour growth was monitored weekly using a Vevo3100 ultrasound imaging system (Fujifilm Visualsonics, The Netherlands). At the end of the experiment, mice were euthanised by excess of $CO_2$ and tumours were collected. Tumour material was further fixed in 10% formalin for histological procedures or snap-frozen in liquid nitrogen.

**Raman spectroscopy analysis.** Raman spectra were acquired on a Renishaw inVia Raman microscope equipped with a 532 nm Nd:YAG laser giving a maximum power of 500 mW, 1800 l mm$^{-1}$ grating, and a Nikon NIR Apo 60×/1.0 N.A. water dipping objective. Tissue sections were dewaxed immediately prior to Raman measurements in xylene (2 × 15 min), 100% ethanol (5 min), 95% EtOH (5 min) and 90% EtOH (5 min). The tissue sections were mapped using a water dipping objective, using a step size of 100 μm in $x$ and $y$, with 1 s acquisition time, 100% laser power and a spectral center of 3000 cm$^{-1}$.

Renishaw Wire 4.1 was used to perform basic pre-processing steps using the inbuilt functions in the software. Cosmic rays were removed followed by baseline subtraction. Baseline was subtracted using the in-built baseline subtraction intelligent fitting function (with an 11th order polynomial fitting and noise tolerance set to 1.50 which was applied to the whole spectral dataset). Custom MATLAB® scripts were then used to perform further analysis. Outlier spectra (high intensity due to saturation or fluorescence) were removed using a threshold function. Spectra were cut to between 2800 and 3020 cm$^{-1}$. Tissue regions were then selected based on the total spectral intensity for the map and all associated spectra were extracted and min–max scaled for comparison between conditions. The resultant spectral data set for each tissue map went through further quality control steps; firstly, excluding spectra with a total spectral intensity less than 60,000, followed by removing spectra out with one standard deviation of the mean. Spectra were scaled to the peak at 2933 cm$^{-1}$, and spectral data sets for all CTL and DECR1 KO samples were combined, with average for each condition plotted for comparison. The ratio of the intensities of the peaks at 2845 and 2935 cm$^{-1}$ was determined for each spectral data point, as was the ratio of the intensities of the peaks at 2880 and 2935 cm$^{-1}$. These values were used to create bar charts depicting the mean and standard deviation values in GraphPad Prism 7. GraphPad Prism 7 was used to perform a Mann–Whitney test to compare CTL to DECR1 KO for each ratio.

**Patient material and immunohistochemistry.** This study was approved by the West of Scotland Research Ethics Committee (05/S0704/94). All subjects provided an informed consent and all experiments conformed to the principles set out in the WMA Declaration of Helsinki and the Department of Health and Human Services Belmont Report.

Immunohistochemical (IHC) staining for DECR1 was performed on 4 μm formalin-fixed paraffin-embedded sections which had previously been incubated at 60 °C for 2 h. The IHC staining was performed on an Agilent Autostainer link 48 (Agilent, Santa Clara, CA, USA).

The sections underwent manual dewaxing through xylene, graded alcohol and then washed in tap water before undergoing heat-induced epitope retrieval (HIER). HIER was performed on an Agilent PT module where the 4 μm sections were heated to 98 °C for 25 min in PT module 1 buffer (Thermo Fisher Scientific, Waltham, MA, USA). After epitope retrieval sections were rinsed in Tris Buffered saline with Tween (Tbt) prior to being loaded onto the autostainer. The sections then underwent peroxidase blocking (Agilent, Santa Clara, CA, USA), washed in Tbt before application of DECR1 antibody (Abcam, Cambridge, UK) at 1/1000 dilution for 40 min. The sections were then washed in Tbt before application of rabbit EnVision (Agilent, Santa Clara, CA, USA) secondary antibody for 35 min. Sections were rinsed in Tbt before applying Liquid DAB (Agilent, Santa Clara, CA, USA) for 10 min. The sections were then washed in water, counterstained with hematoxylin and mounted using DPX (Thermo Fisher Scientific, Waltham, MA, USA).

**siRNA transfection.** Cells were seeded in 6-well plates to reach 70% confluence and allowed to attach overnight. ON-TARGETplus smartpool siRNAs against AMPKa (L-005027-00), AR (L-003400-00), DECR1 (L-009642-00-005), as well as non-targeting siRNA (D-001810-01-20) were purchased from Dharmacon (Dharmacon, Horizon inspired cell solutions, Cambridge, UK). DECR1 over-expressing plasmid was purchased from Origene (RC211257, Cambridge Bioscience, Cambridge, UK). Transfections were performed using Lipofectamine RNAimax (Invitrogen, Thermo Fisher Scientific, Waltham, MA, USA) according to the manufacturer's protocol.

**Generation of DECR1 stable KO cells.** DECR1 deletion was performed using the CRISPR CAS9 gene editing technology. Briefly 1 × 10$^6$ cells were transfected with commercially available DECR1 KO or CTL plasmids (Santa Cruz Technologies, Dallas, TX, USA) using a nucleofector (Amaxa Biosystems, Lonza, Basel, Switzerland) according to the manufacturer's instructions. Cells were then cultured in the presence of 1 μg ml$^{-1}$ of puromycin (Sigma Aldrich, St Louis, MI, USA) to

allow for clonal selection. Final knock-out of the protein was confirmed using western blot.

**Cell proliferation.** 6–8 × 10$^5$ cells were seeded in 6-well plates and allowed to attach for 16 h. The next day, cells were either directly harvested with trypsin (T0) or allowed to grow for additional 48 or 72 h. Cells were then harvested, suspended in a defined volume of serum-containing medium and counted using a CASY cell counter (Roche, Basel, Switzerland). Final cell number was normalised to the initial cell count obtained at T0.

**qPCR analysis.** RNA was extracted from cells (70–80% confluence) using the RNeasy Mini Kit (Qiagen, Hilden, Germany) with on-column DNase digestion (RNase-Free DNase Set, Qiagen, Hilden, Germany). cDNA was prepared from 4 μg RNA using High-Capacity cDNA Reverse Transcription Kit (Thermo Fisher Scientific, Waltham, MA, USA). Real-time PCR was performed using TaqMan Universal Master Mix (Thermo Fisher Scientific, Waltham, MA, USA) with primer-appropriate Universal ProbeLibrary probes (Roche, Basel, Switzerland) and the ABI 7500 FAST qPCR system (Thermo Fisher Scientific, Waltham, MA, USA). Gene expression was normalised to the reference CASC3 gene, and is shown relative to levels in control cells. A list of the primers (Thermo Fisher Scientific, Waltham, MA, USA) used in this study is provided in Supplementary Table 1.

**Immunoblotting.** Cells or crushed frozen tumour tissues were lysed in SDS buffer (1% SDS supplemented with protease and phosphatase inhibitors) and protein concentration was determined using the BCA protein assay kit (Thermo Fisher Scientific, Waltham, MA, USA). Twenty microgram of proteins were then loaded on to a 4–12% gradient SDS-PAGE gel (Invitrogen, Thermo Fisher Scientific, Waltham, MA, USA) and transferred to a PVDF membrane (GE Healthcare, Chicago, IL, USA). Membrane was blocked in 5% milk-TBST and probed overnight with primary antibodies (see Supplementary Table 2) diluted in 5% BSA-TBST. The membrane was then washed three times with TBST, incubated with respective HRP-conjugated secondary antibodies diluted in 5% milk-TBST, washed another three times with TBST and revealed using the ECL kit (GE Healthcare, Chicago, IL, USA). Images were acquired on a MyECL machine (Thermo Fisher Scientific, Waltham, MA, USA).

**Immunofluorescence.** Cells were seeded on coverslips in 24-well plates to reach 50% confluence and allowed to attach overnight. Cells were next fixed in ice-cold Methanol/Acetone buffer, washed three times with PBS-Tween, blocked for 30 min in a PBS/BSA (5%) solution and probed with primary antibody overnight (anti AR, sc-816, Santa Cruz Technologies, Dallas, TX, USA). The next day, coverslips were washed three times with PBS-Tween, incubated with fluorophore-coupled secondary antibodies (Abcam, Cambridge, UK) and washed again three times with PBS. Coverslips were mounted using Diamond Prolong with DAPI (Thermo Fisher Scientific, Waltham, MA, USA). Pictures were taken on a Nikon A1R confocal microscope (Nikon Instruments Europe B.V., Amsterdam, The Netherlands).

**Chromatin immunoprecipitation (ChIP).** Chromatin was prepared with the tru-ChIP™ Chromatin Shearing Kit (Covaris, Brighton, UK) according to manufacturer's instructions. Each sample was sonicated for 10 min using Covaris sonicator. ChIP were performed using the IP-Star Compact Automated System (Diagenode, Liege, Belgium). Briefly, 4 μg of isolated chromatin was immunoprecipitated with either 1 μg ChIP grade antibody (anti-AR 17-10489, Millipore Burlington, MA, USA or anti-AR #5153, Cell Signalling Technology, London, UK) or 1 μg of IgG (C15410206, Diagenode, Liege, Belgium) in dilution buffer (0.01% SDS; 1.1% Triton X 100; 2 mM EDTA; 16.7 mM Tris-Cl pH 8.0; 167 mM NaCl; 1× protease inhibitor cocktail, Sigma Aldrich, St Louis, MI, USA). The DNA/protein complexes were washed four times in IP Wash buffer (100 mM Tris-HCl pH 8.0; 500 mM LiCl 1%; Triton X100; 1% deoxycholic acid. After reversal of crosslinking, the immunoprecipitated DNA was purified by a regular DNA extraction protocol and analysed employing RT-qPCR with the SYBR-Green Takara (Ozyme, Paris, France) and step one plus applied Real-Time PCR system. The PCR conditions were 10 min at 95 °C followed by 45 cycles of 10 s at 95 °C, 30 s at 60 °C, and 30 s at 72 °C. The following, primers were used: promoter KLK3 5′ GCC TGG ATC TGA GAG AGA TAT CAT C 3′ [R] and 5′ ACA CCT TTT TTT TTC TGG ATT GTT G 3′ [F]; and promoter FKBP5 5′ GCA TGG TTT AGG GGT TCT TGC 3′ [R] and 5′ AAC ACC CTG TTC TGA ATG TGG C 3′ [F].

**Statistical analysis.** Statistical analyses were performed using GraphPad PRISM software v7.05 (GraphPad Software Inc, San Diego, CA, USA).

**Data reproducibility.** Figure 1: Panels a, b, f, h: representative image from three independent biological experiments. Panel c (top): $n = 6$ independent biological experiments. Panel c (bottom): $n = 55$ (LN), 40 (BIC), 51 (APA), 51 (ENZ) organoids examined over two independent biological experiments. Panels d, e: $n = 3$ independent biological experiments. Panel g: $n = 9$ (three independent biological experiments performed in triplicates).

Figure 2: Panel b: $n = 3$ independent biological experiments; Panels c, d, e: $n = 3$ independent wells from the same cell culture.

Figure 3: Panel a: $n = 3$ independent wells from the same cell culture. Panel b: representative image from three independent biological experiments. Panel c: $n = 3$ independent biological experiments.

Figure 4: Panel a: $n = 3$ independent wells from the same cell culture. Panel b: representative image from three independent biological experiments.

Figure 5: Panels b, c, f: representative image from three independent biological experiments. Panel d: $n = 9$ (three independent biological experiments performed in triplicates). Panel e: $n = 6$ (three independent biological experiments performed in duplicates). Panel g: $n = 14$.

Figure 6: Panels a, g, h: representative image from three independent biological experiments. Panels b, i, j: $n = 6$ (three independent biological experiments performed in duplicates). Panels c, d: $n = 3$ independent biological experiments. Panels e, f: $n = 3$ independent wells from the same cell culture.

Figure 7: Panel a: $n = 6$ mice per group. Panel b: $n = 5$ mice per group. Panel c: $n = 7203$ (CTL), 5425 (KO) peak intensities that were extracted from four mice per group. Panel d: $n = 3$ mice per group.

**Reporting summary**. Further information on research design is available in the Nature Research Reporting Summary linked to this article.

## Data availability

The raw files and the MaxQuant search results files have been deposited as partial submission to the ProteomeXchange Consortium via the PRIDE partner repository [Perez-Riverol Y, Csordas A, Bai J, Bernal-Llinares M, Hewapathirana S, Kundu DJ, Inuganti A, Griss J, Mayer G, Eisenacher M, Pérez E, Uszkoreit J, Pfeuffer J, Sachsenberg T, Yilmaz S, Tiwary S, Cox J, Audain E, Walzer M, Jarnuczak AF, Ternent T, Brazma A, Vizcaíno JA (2019) The PRIDE database and related tools and resources in 2019: improving support for quantification data. Nucleic Acids Res 47(D1):D442-D450 (PubMed ID: 30395289)] with the dataset identifier PXD016836. The following databases were used in this study: The Cancer Genome Atlas (TCGA—https://tcga-data.nci.nih.gov/tcga/); GSE21034; STRING v11.0 (https://string-db.org/cgi/input.pl). All the data supporting the findings of this study are available within the article and its supplementary information files and from the corresponding author upon reasonable request. A reporting summary for this article is available as Supplementary Information file. The source data underlying Figs. 1–7 and Supplementary Figs. 1–6 are provided as a Source data file. Raw data for lipidomics experiment are provided as Supplementary Data 2.

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

## Acknowledgements

This work was supported by Cancer Research UK Beatson Institute core funding (C596/A17196) and CRUK core group awarded to HYL (A15151) and to SZ (A12935). P.P. and E.H. were funded by grants from "La ligue Contre le Cancer", "la région Bourgogne Franche-Comté" and "Canceropole Grand Est". M.S. is a Medical Research Council Clinical Research Fellow (MR/L017997/1). C.N. is the recipient of CRUK Clinical Research Fellowship (grant 300444-01). D.G. and K.F. acknowledge support from the EPSRC grant EP/L014165/1 that supported L.J. S.-M.F. acknowledges FWO funding and KU Leuven Methusalem co-funding.

## Author contributions

A.B. and H.Y.L. designed the study. A.B., C.F., E.M., R.P., L.J., M.P., G.H.M., P.P., E.H., C.Ni, M.S., E.Ma, D.S., G.R.B. and S.L. performed the experiments. A.B., C.N., L.J., M.P., G.H.M., P.P., E.Ma and P.R. analysed the data. A.B., R.P., L.J., G.H.M., P.P., L.G., J.K., D.G., K.F., G.M.M., S.-M.F., S.Z. and H.Y.L. interpreted and discussed the data. A.B. and H.Y.L. wrote the manuscript. All authors critically reviewed the manuscript.

## Competing interests

The authors declare no competing interests.
