## [Peer Review File · Nature Communications]

Reviewers' comments:

Reviewer #1 (Remarks to the Author):

The data described in this manuscript implicate DECR1, an enzyme involved in fatty acid degradation in androgen-resistance in prostate cancer PrCa. The authors have generated three independent treatment-resistant PrCa cell lines using different AR inhibitors (ARi). They then use a proteomics and metabolomics approach to identify commonly deregulated pathways. Among the deregulated pathways was mitochondrial metabolism, indicating that ARi alter the metabolism of PrCa cells. In addition, metabolomic and expression profiling indicated a shift to increased glucose consumption.

They also find that DECR1, an enzyme involved in the degradation of poly-unsaturated fatty acids, is strongly upregulated in androgen insensitive or castration resistant xenograft tumours and patient samples. They show that CRISPR-mediated deletion of DECR1 alters cellular metabolism in PrCa cells and results in a change in lipid composition, with an accumulation of poly-unsaturated species. The authors then speculate that this could increase lipid peroxidation, leaving cells more sensitive to ferroptosis. Finally, the authors show that deletion of DECR1 reduces cell proliferation in vitro and attenuates tumour growth in vivo.

Overall, this study shows an interesting upregulation of DECR1 in aggressive prostate cancer. However, the mechanistic analysis of the role of DECR1 in promoting cancer cell survival is quite preliminary. In particular, the link to ferroptosis is not supported very well by the data. There are also a number of issues with the presentation of the data that need to be addressed before the manuscript can be considered for publication.

Major comments:

- While the proteomic and metabolomic analysis of ARi resistant cells is interesting, many of the indicated pathways are not further explored. For example, what is the significance of increased mitochondrial metabolism for the ARi resistance? What are the consequences of the alterations in glucose metabolism and fatty acid synthesis? This part of the manuscript seems very descriptive without providing a clear mechanism. This also applies to the metabolomics analysis in the DHCR1 ko cells, which does not contribute to the conclusions regarding PUFA metabolism.

- The change in lipidomic profile in DHCR1 ko cells is intriguing but it is not clear whether this is caused by altered FA metabolism or changes in FA uptake vs de novo synthesis. Also, are there corresponding changes in desaturation profiles between ARi sensitive and resistant cells? Would overexpression of DHCR1 induce ARi resistance?

- The connection between DHCR1 upregulation and ferroptosis resistance is correlative at best. The authors need to show that cells die indeed of ferroptosis, by showing increased lipid peroxidation and rescue with lipid-specific antioxidants.

- Finally, as DHCR1 ko cells show reduced proliferation already in vitro, it is no great surprise that they also show a reduced ability for tumour growth. The analysis of lipids using Raman spectroscopy does not provide any mechanistic insight into potential accumulation of poly-unsaturated lipid species, which could result in enhanced lipid peroxidation and cell death.

Specific points:

1) Fig. 1A-C: The authors need to quantify cell size and spheroid sizes in order to draw these conclusions.

2) The signal of nuclear AR in Fig. 1H looks quite different in Apalut-resistant cell lines. Were these images taken at the same settings? Is it possible to quantify?

2) It is unclear whether the labelling experiments were conducted at dynamic or steady state. This is important for the conclusions that can be drawn from the data. Different metabolites reach steady state labelling at different times. Without this knowledge initially established using different labelling times, it is difficult to conclude whether a pathway is more active or whether the proportional labelling of the metabolite from the labelled precursor is changed.

3) The WB in Fig. S2A needs to be quantified.

4) Many important results are in the supplementary figures, making it very hard to follow the argument. For example, the increased glutathione synthesis from glucose (Fig. S3c) should be shown in main figures.

5) In Fig. S4A, the P-AMPK signal seem to mirror that of AMPK. This does not indicate AMPK activation. The authors could check P-ACC as AMPK substrate. Clearly, AR regulates AMPK expression rather than activation. Which AMPK subunit was detected here?

6) The diagram in Fig. 4a is very unclear. What is in brackets? Why are the protein names not stated here? The 8 (5) proteins deregulated in all resistant cell lines should be named. Why are there 13 candidates as mentioned in the text?

7) The WB in Fig. 4C should also be quantified to allow strong conclusions about the upregulation of DECR1 after ARI-treatment.

8) Is the effect of silencing of DECR1 and enzalutamide treatment shown in Fig. 4D synergistic or just additive?

9) No information on patient numbers provided in Fig. S5G.

10) Are the cell lines shown in Fig. 5a-b really isogenic? How were they generated?

11) Is the reduced proliferation of DECR1 ko cells rescued by addition of androgen? This is implied in the text.

12) Where are the TCA cycle metabolites detected in Fig. S6a?

13) Given that DECR1 ko cells already show reduced proliferation (Fig. 5D), the lower cell number in presence of the GPX4 inhibitor, given relative to cell number at beginning of experiment as stated in the legend, does not indicate increased dependence on removal of lipid peroxides. The reference to figure 5h is missing.

Reviewer #2 (Remarks to the Author):

This is a well written and conducted study about metabolic and proteomic adaptations in prostate cancer cells. The findings are possibly worthwhile publishing. However, I have some comments which should be addressed. The dependency of cancer cells on de novo lipid biosynthesis is not new, otherwise FASN inhibitors would not have been investigated as promising drug candidates since the late 90s. Moreover, the toxic effect of PUFA on cancer cells is another well-known fact. However, the here presented study lays a novel link between ARI resistance and important metabolic alterations.

I appreciated the Raman approach, nice idea for looking into lipid content in tumor biopsies.

Major:

The authors' findings are intriguing. However, it would have been very interesting to better understand how DECR1 influences PUFA metabolism. Out of several reasons: recently several

reports point out how PUFA derived mediators might play important roles in cancer (e.g. Proc Natl Acad Sci U S A. 2019 Mar 26; 116(13): 6292–6297). In turn, it would have been an intriguing link to at least discuss this in a bit more detail. Moreover, as lipid homeostasis is heavily controlled by LXR I was wondering if there is a link to this transcription factor? LXR has taken center stage in lipid homeostasis not only in immune cells but also during the development of cancer e.g. Uncoupling Nuclear Receptor LXR and Cholesterol Metabolism in Cancer. Fabiola Bovenga et al. DOI: <https://doi.org/10.1016/j.cmet.2015.03.002> Cell Metab. It would be interesting to see whether LXR might not be influenced under ARI (or vice versa KO) as well? This would shed an entirely different light on the reported findings about PUFA metabolism and maybe integrate nicely with DCER1.

Moreover, the given proof for DCER1 influencing PUFA degradation is mainly based on lipidomics analysis where no internal standards have been used. Frankly, this is not enough to convince me. This is a key message of the paper and this should be better investigated. The authors could simply hydrolyze lipids and carry out fatty acid compositional analysis. Or even better, do it with and without hydrolysis, possibly also shedding light on free PUFA versus total PUFA, as bound PUFA are usually not harmful to cancer cells, while free PUFA are. In turn, I miss the more thorough investigation of 2 key points, PUFA content using a really quantitative method, not the lipidomics one described in the manuscript, and secondly free versus bound PUFA as this is key for PUFA toxicity.

I wonder how lipids were quantified? No internal standards are specified and using RP-18 separation it is well-known that PE overlap with PC, making PE quantification quite tricky at times. Could the authors please comment?

Figure 4f is very hard to grasp. Frankly, the two pictures look entirely different and I am not sure what I am really evaluating here?

Minor

Times sign and letter x should not be interchanged

Reviewer #3 (Remarks to the Author):

Minor comments:

Can the authors provide details on the specific data processing performed in Renishaw Wire and what sort of custom baseline subtraction was used?

What was the actual laser power on the sample? The laser offers 500 mW (which is relatively high compared to most studies) and 100% laser power was used.

It would be good for non-experts to know why Raman is a tool of choice for this specific application. i.e., the authors should mention the high sensitivity to lipids.

On page 12-13 the authors should either acquire a Raman spectrum from cholesterol or provide a citation for cholesterol Raman spectra and lipids in general.

Reviewers' comments:

Reviewer #1 (Remarks to the Author):

The data described in this manuscript implicate DECR1, an enzyme involved in fatty acid degradation in androgen-resistance in prostate cancer PrCa. The authors have generated three independent treatment-resistant PrCa cell lines using different AR inhibitors (ARi). They then use a proteomics and metabolomics approach to identify commonly deregulated pathways. Among the deregulated pathways was mitochondrial metabolism, indicating that ARi alter the metabolism of PrCa cells. In addition, metabolomic and expression profiling indicated a shift to increased glucose consumption. They also find that DECR1, an enzyme involved in the degradation of poly-unsaturated fatty acids, is strongly upregulated in androgen insensitive or castration resistant xenograft tumours and patient samples. They show that CRISPR-mediated deletion of DECR1 alters cellular metabolism in PrCa cells and results in a change in lipid composition, with an accumulation of poly-unsaturated species. The authors then speculate that this could increase lipid peroxidation, leaving cells more sensitive to ferroptosis. Finally, the authors show that deletion of DECR1 reduces cell proliferation in vitro and attenuates tumour growth in vivo.

Overall, this study shows an interesting upregulation of DECR1 in aggressive prostate cancer. However, the mechanistic analysis of the role of DECR1 in promoting cancer cell survival is quite preliminary. In particular, the link to ferroptosis is not supported very well by the data. There are also a number of issues with the presentation of the data that need to be addressed before the manuscript can be considered for publication.

We thank the Reviewer for finding our story of interest and for the wise suggestions. We have now performed several experiments to address each of these concerns and we feel that they have contributed to significantly improve our manuscript.

Major comments:

- While the proteomic and metabolomic analysis of ARi resistant cells is interesting, many of the indicated pathways are not further explored. For example, what is the significance of increased mitochondrial metabolism for the ARi resistance? What are the consequences of the alterations in glucose metabolism and fatty acid synthesis? This part of the manuscript seems very descriptive without providing a clear mechanism.

We agree with the Reviewer that the original story was lacking a clear link between the metabolic phenotype of the ARi-resistant cells and the role of DECR1 in lipid homeostasis. Thanks to the Reviewers' comments, we have now addressed this issue by characterising the cellular lipidome of the resistant cells. Indeed, we had initially demonstrated that ARi-resistant cells switched their metabolism towards increased glucose utilisation, mainly to be used for the generation of fatty acids (FA), in an AR-dependent manner (Figure 2, 3 and 4). We have now demonstrated that this metabolic adaptation ultimately leads to a dramatic reorganisation of the cellular lipidome of the ARi-resistant cells (Figure 3). These cells exhibit a highly dysregulated lipid metabolism, accumulating large amounts of multiple lipid molecules, particularly polyunsaturated triglycerides and sphingolipids. In this context, increased mitochondrial metabolism would be beneficial not only to support FA and lipid synthesis, via the generation of citrate and acetyl-CoA, but also to regulate lipid homeostasis, for example through an increase in β -oxidation. This hypothesis is further supported by the high

expression of several transcription factors involved in mitochondrial metabolism and lipid homeostasis, such as PGC1 α , PPAR γ and LXR, in ARI-resistant cells. Along the same line, overexpression of DECRI, through its role on regulating lipid saturation levels (Figure 6), is necessary to maintain lipid homeostasis in the metabolically-rewired resistant cells. We have now discussed this part in the appropriate section.

Understanding the detailed molecular mechanisms of how mitochondrial metabolism affects ARI resistance is an interesting subject; however, we feel that this represents an entirely new project, which goes beyond the scope of the current study. By providing all our proteomic data (multiple comparisons in 2D and 3D cultures), we hope that we and/or others will be able to address that aspect in the future.

This also applies to the metabolomics analysis in the DHCR1 ko cells, which does not contribute to the conclusions regarding PUFA metabolism.

We agree with this comment and have decided to focus on the lipid/fatty acid aspect of DECRI deletion (Figure 6). Therefore the metabolomics experiment on DECRI deletion has now been moved to supplementary data (Supplementary Figure 6c).

- The change in lipidomic profile in DHCR1 ko cells is intriguing but it is not clear whether this is caused by altered FA metabolism or changes in FA uptake vs de novo synthesis. Also, are there corresponding changes in desaturation profiles between ARi sensitive and resistant cells? Would overexpression of DHCR1 induce ARi resistance?

We thank the Reviewer for this comment. As mentioned above, we have now performed global lipidomic profiling of the ARI-resistant cells and demonstrated that these cells accumulate large amounts of lipids, especially polyunsaturated triglycerides and sphingolipids (Figure 3c). This result is consistent not only with the cellular morphology (larger/flatter cells than WT LNCaP) and the metabolic rewiring observed in the resistant cells, but also with elevated levels of DECRI and other lipid-related enzymes that are needed to balance lipid homeostasis in these cells (Figure 2a).

We have monitored fatty acid synthesis, derived from labelled glucose, in DECRI-deficient cells and did not find any difference when compared to CTL cells (Supplementary Figure 6f). In addition, a recent proteomic analysis of the DECRI KO cells performed in our lab revealed that the expression levels of the several lipid transporters detected were not altered in these cells when compared to CTL (see graph below, BIP is shown as positive control - data not included in this manuscript). Collectively, our data suggest that the changes observed in the lipid composition of the cells lacking DECRI probably result from loss of enzymatic function.

Finally, we tested whether *DECRI* overexpression would be sufficient to induce ARI resistance by inducing *DECRI* in WT LNCaP. Transient (48h) overexpression of *DECRI* marginally increased cell proliferation in normal culture conditions and upon enzalutamide treatment (Figure 5e). While we cannot exclude that *DECRI* overexpression might be important for other cancer-related processes (such as migration/invasion), our hypothesis is that *DECRI* overexpression in prostate cancer is more likely to be part of a long term, adaptive, metabolic response, rather than an isolated event. This is supported by the fact that multiple lipid metabolism-related/redox enzymes are overexpressed in ARI-resistant cells. From these enzymes, *DECRI* was the most consistently up-regulated and the one showing the highest clinical relevance in patient materials.

- The connection between *DHCR1* upregulation and ferroptosis resistance is correlative at best. The authors need to show that cells die indeed of ferroptosis, by showing increased lipid peroxidation and rescue with lipid-specific antioxidants.

We thank the Reviewer for raising this important issue and for the helpful suggestion. We have now used both Trolox and liproxstatin, two inhibitors of ferroptosis, to rescue the phenotype that was observed in DECRI-deficient cells upon RSL3 treatment. Both inhibitors were not only able to reverse the RSL3-mediated effects observed on cell proliferation and cell morphology of the DECRI-deficient cells (Figure 6j and Supplementary Figure 6g), but also to abolish the anti-proliferative effect of DECRI silencing under these conditions. This result reinforces the idea that DECRI plays a protective role in the induction of ferroptosis in prostate cancer.

- Finally, as *DHCR1* ko cells show reduced proliferation already in vitro, it is no great surprise that they also show a reduced ability for tumour growth. The analysis of lipids using Raman spectroscopy does not provide any mechanistic insight into potential accumulation of poly-unsaturated lipid species, which could result in enhanced lipid peroxidation and cell death.

We decided to use Raman spectroscopy as this technique allowed us to evaluate intra-tumoural lipid and cholesterol content on fixed samples, in a label free and non-destructive manner. Therefore, we think that this technique bears high potential to be used in clinical samples/biopsies in the future. However, we agree that Raman spectroscopy did not directly support the determination of the lipid classes altered upon DECRI KO. To partially answer this question, we performed another LC-MS-based lipidomic analysis on frozen CTL (control) and DECRI KO tumours. Even with the limitations of performing lipidomics on whole tumour extracts (including higher variability between samples, presence of stroma), we could show that DECRI KO tumours displayed lower levels of certain classes of abundant triglycerides, which accounted for the main differences between the two conditions. These cells further accumulated several species of ceramides and polyunsaturated phospholipids, a result that was already observed in vitro and has been linked to mitochondrial dysfunction and cell death (Kim, et al. Neurochem. Res. 2005. 30 (8),969-979 and Yang, et al. PNAS. 2016. 113, E4966-4975).

Specific points:

1) Fig. 1A-C: The authors need to quantify cell size to and spheroid sizes in order to draw these conclusions.

Cell and spheroid sizes have now been quantified; the results are depicted in Figure 1c.

2) The signal of nuclear AR in Fig. 1H looks quite different in Apalut-resistant cell lines. Were these images taken at the same settings? Is it possible to quantify?

Initial immunofluorescence images were taken at the same settings. However, the original aim for this experiment was to show whether nuclear localisation of the receptor can still occur in resistant cells. Therefore we only showed representative pictures of a nuclear AR staining in the different conditions. Quantification of total AR protein between the different cell lines can be seen in Figure 1f and we have now performed additional WB of nuclear/cytoplasmic extracts to show quantitative expression of nuclear AR across the different cell lines (Supplementary Figure 1b).

2) It is unclear whether the labelling experiments were conducted at dynamic or steady state. This is important for the conclusions that can be drawn from the data. Different metabolites reach steady state labelling at different times. Without this knowledge initially established using different labelling times, it is difficult to conclude whether a pathway is more active or whether the proportional labelling of the metabolite from the labelled precursor is changed.

We agree with Reviewer 1 that different metabolites reach steady state labelling at different times. The initial tracing experiments were performed after 1 and 24 hours of incubation with labelled glucose. In both experiments, enrichment of glycolytic intermediates was the main observed result, and labelling of the majority of these intermediates was maximal after 1 h (Figure 2c – G6P, F16P, DHAP, G3P, 2PG, PEP). These effects were maintained after 24 h, suggesting a global increase in glycolysis in all three resistant cells. We have now added examples for labelled pyruvate and phosphoenolpyruvate levels after 24 h and modified the text accordingly (Figure 2d). After 24 h of incubation, we also observed a consistent increase in the level of labelled GSH, which was indicative of increased redox stress. These data are now presented in Figure 2e. By contrast, intermediates of the TCA cycle or from the PPP were not consistently enriched in all three resistant cells and were not discussed further in this manuscript.

3) The WB in Fig. S2A needs to be quantified.

WB from 3 biological replicates have been quantified and data are now included in Supplementary Figure 2e. Moreover, we have decided to remove the PKM1/2 panel as the average overexpression of the protein was less than 15% in the Bicalutamide-resistant cell line.

4) Many important results are in the supplementary figures, making it very hard to follow the argument. For example, the increased glutathione synthesis from glucose (Fig. S3c) should be shown in main figures.

We thank the Reviewer for this comment. This panel (original Supplementary Figure 3c) is now presented in Figure 2e. We have also split original Figure 2 into two: the new Figure 2 focuses only on glucose metabolism while we created a new Figure 3 to focus on the lipid metabolism of ARI-resistant cells. This allowed us to show more pertinent data in the main manuscript.

5) In Fig. S4A, the P-AMPK signal seem to mirror that of AMPK. This does not indicate AMPK activation. The authors could check P-ACC as AMPK substrate. Clearly, AR regulates AMPK expression rather than activation. Which AMPK subunit was detected here?

We focused mainly on AMPK α . We agree with Reviewer 1 that AR seems to regulate AMPK α expression. We have now checked ACC phosphorylation, as suggested, and observed that ACC was strongly phosphorylated (Supplementary Figure 4a) in all three resistant cell lines. This suggested to us that both AMPK α expression and activation occurred in ARI resistant cells. We have now modified the figures and the text accordingly.

6) The diagram in Fig. 4a is very unclear. What is in brackets? Why are the protein names not stated here? The 8 (5) proteins deregulated in all resistant cell lines should be named. Why are there 13 candidates as mentioned in the text?

We apologise for this lack of clarity and we thank the Reviewer for raising this issue. As mentioned in the legend, the free numbers represent the significantly up-regulated proteins in a specific condition (both 2D and 3D), while the numbers that were put into brackets represent the significantly down-regulated proteins. Therefore the 13 dysregulated proteins represented both the up- and down-regulated proteins (8 up + 5 down) and were given in Supplementary Table 2. Following Reviewer 1's advice, we have now modified the figure to better explain the signification of these numbers. We have also named the members of our proteomic signature directly in Figure 5a.

7) The WB in Fig. 4C should also be quantified to allow strong conclusions about the upregulation of DECR1 after ARI-treatment.

WB have been quantified and data are now included in Figure 5c.

8) Is the effect of silencing of DECR1 and enzalutamide treatment shown in Fig. 4D synergistic or just additive?

At this concentration of enzalutamide, the effect DECR1 silencing is additive and not synergistic.

9) No information on patient numbers provided in Fig. S5G.

We thank the Reviewer for pointing this out. Patient numbers have now been added in Supplementary Figure 5g.

10) Are the cell lines shown in Fig. 5a-b really isogenic? How were they generated?

LNCaP AI (androgen insensitive) are derived from WT LNCaP cells that have been long-term cultured in medium lacking androgens (supplemented with Charcoal-stripped serum instead of regular FBS) (Yu, et al. Int. J. Mol. Med. 2017. 40(5), 1426-1434). LNCaP AI cells (as well as Bicalutamide/Abiraterone/Enzalutamide-resistant cells) have further been authenticated using genomic DNA and show >95% genetic identity to WT LNCaP.

11) Is the reduced proliferation of DECR1 ko cells rescued by addition of androgen? This is implied in the text.

We thank the Reviewer for this comment and apologise for the confusion. We wrote “Loss of DECRI expression impaired in vitro cellular proliferation of LNCaP AI cells in androgen-depleted conditions by an average of 32%” because LNCaP AI cells are routinely cultured in androgen-depleted conditions (CSS medium). However LNCaP AI cells are not responsive, in terms of proliferation, to the addition of androgens (such as DHT) and we have now modified the sentence to avoid this confusion. Nevertheless, we tested whether DHT treatment could influence the proliferation of DECRI KO cells and, as expected, we did not observe any positive effect of DHT treatment on the proliferation of CTL nor DECRI KO LNCaP AI cells (see graph below).

12) Where are the TCA cycle metabolites detected in Fig. S6a?

We thank the Reviewer for this comment and again apologise for the confusion. The initial sentence “We did not observe any significant changes in the expression of TCA cycle metabolites or in the several carnitine derivatives that were identified (Suppl. Fig. 6a)” was misleading and we have modified the figure in order to show the TCA cycles metabolites that were detected in the analysis (Supplementary Figure 6d).

13) Given that DECRI ko cells already show reduced proliferation (Fig. 5D), the lower cell number in presence of the GPX4 inhibitor, given relative to cell number at beginning of experiment as stated in the legend, does not indicate increased dependence on removal of lipid peroxides.

We agree with the Reviewer that the DECRI KO cells proliferate less than control cells in absence of any treatment. However, we observed that RSL3 treatment selectively killed DECRI KO cells rather than just slowing down their proliferation. Indeed, **after 48 h**, DECRI deletion reduces cell proliferation by ~20% when compared to CTL cells. **Upon RSL3 treatment, at the same timepoint (48 h)**, proliferation of the DECRI KO cells was reduced by ~45% compared to CTL cells. Figure 6b might be misleading as it shows a ~30% decrease **after 72 h**, but RSL3 treatment was performed for only 48 h. In addition to this, the change in cell morphology (round cells) that was observed in the DECRI KO cells upon RSL3 treatment (by comparison to CTL cells) indicated that these cells underwent cell death more than just reduced proliferation. To make this point clear, we have now included pictures of CTL and DECRI KO cells upon RSL3 treatment in Supplementary Figure 6g. Moreover, as rightly suggested by the Reviewer, we were able to rescue the RSL3-dependent reduction in cell number observed in DECRI KO cells by using two ferroptosis inhibitors (Trolox and liproxstatin). Data are now presented in Figure 6j.

The reference to figure 5h is missing.

We have now added the reference to original Figure 5h (which is now Figure 6h).

Reviewer #2 (Remarks to the Author):

This is a well written and conducted study about metabolic and proteomic adaptations in prostate cancer cells. The findings are possibly worthwhile publishing. However, I have some comments which should be addressed. The dependency of cancer cells on de novo lipid biosynthesis is not new, otherwise FASN inhibitors would not have been investigated as promising drug candidates since the late 90s. Moreover, the toxic effect of PUFA on cancer cells is another well-known fact. However, the here presented study lays a novel link between ARI resistance and important metabolic alterations. I appreciated the Raman approach, nice idea for looking into lipid content in tumor biopsies.

We thank the Reviewer for showing interest in our story and for the favourable comments.

Major:

The authors' findings are intriguing. However, it would have been very interesting to better understand how DECR1 influences PUFA metabolism. Out of several reasons: recently several reports point out how PUFA derived mediators might play important roles in cancer (e.g. Proc Natl Acad Sci U S A. 2019 Mar 26; 116(13): 6292–6297). In turn, it would have been an intriguing link to at least discuss this in a bit more detail. Moreover, as lipid homeostasis is heavily controlled by LXR I was wondering if there is a link to this transcription factor? LXR has taken center stage in lipid homeostasis not only in immune cells but also during the development of cancer e.g. Uncoupling Nuclear Receptor LXR and Cholesterol Metabolism in Cancer. Fabiola Bovenga et al. DOI:<https://doi.org/10.1016/j.cmet.2015.03.002> Cell Metab. It would be interesting to see whether LXR might not be influenced under ARI (or vice versa KO) as well? This would shed an entirely different light on the reported findings about PUFA metabolism and maybe integrate nicely with DCER1.

We thank the Reviewer for this interesting suggestion. We have checked the expression level of LXR in our resistant cell lines by qPCR. Both α and β isoforms of the receptor (NR1H3 and NR1H2 respectively) were significantly enriched in all three resistant cell lines in comparison to WT LNCaP. These data are now included in Supplementary Figure 2d. However, two well-characterised LXR target genes (ABCA1 and ABCG1) showed opposite patterns of expression among the different ARI cell lines (see figure below), and silencing of LXR α or LXR β did not influence the expression of DECR1 mRNA in LNCaP AI. Therefore, while LXR dysregulation is likely to be involved in the metabolic response of prostate cancer cells to ARI, more work needs to be done to clarify the mechanistic role of LXR in ARI resistance, which goes beyond the scope of the current study.

Moreover, the given proof for DCER1 influencing PUFA degradation is mainly based on lipidomics analysis where no internal standards have been used. Frankly, this is not enough to convince me. This is a key message of the paper and this should be better investigated. The authors could simply hydrolyze lipids and carry out fatty acid compositional analysis. Or even better, do it with and without hydrolysis, possibly also shedding light on free PUFA versus total PUFA, as bound PUFA are usually not harmful to cancer cells, while free PUFA are. In turn, I miss the more thorough investigation of 2 key points, PUFA content using a really quantitative method, not the lipidomics one described in the manuscript, and secondly free versus bound PUFA as this is key for PUFA toxicity.

We would like to thank the Reviewer for this remark and for the useful suggestion. Although our lipidomic analysis has been fully validated in-house (see below), we agree with Reviewer 2 that measuring fatty acid composition in the presence or absence of DCER1 consolidates our report. Therefore, we have performed both total and free fatty acid analysis of CTL and DCER1 KO cells. We found that, in comparison to CTL cells, DCER1-deficient cells accumulated larger amounts of total and free PUFAs (arachidonic and docosahexaenoic acids) while they displayed lower levels of total and free MUFAs (palmitoleic and sapienic acids). These results are in line with the ones obtained from the LC-MS lipidomics analysis, which showed that DCER1-deficient cells accumulated polyunsaturated lipids. Results from TFA and FFA analyses are now presented in Figure 6e and 6f, respectively.

I wonder how lipids were quantified? No internal standards are specified and using RP-18 separation it is well-known that PE overlap with PC, making PE quantification quite tricky at times. Could the authors please comment?

For the global lipidomics analysis we used a relative quantitation method. We used a single phase extraction of cells using a mixture 1:1 butanol:methanol, and we used a QC lipid standard run at the beginning and end of each analytical run to check for any variation in retention time or in accurate mass. In addition, we did spike SPLASH lipidomic mixture (AVANTI) to each sample and 8 standards were detected by the LipiDex software (Supplementary Table 1). We use this internal standard as a QC reference for our method. Overall, the %CV for eight deuterated standards is <15%, indicating the robustness of our analytical methodology. Moreover, following the suggestions of Reviewer 2, both free and total PUFA analysis using a purely quantitative method was performed. Using this method, we observed an increase in the polyunsaturated fatty acids, particularly arachidonic acid and docosahexaenoic acid (DHA), in DCER1 KO cells. Thus, the purely quantitative results validated the relative quantitative lipidomics method.

We used a well-established chromatographic gradient on a C18 column that allows the separation of the major lipid classes (i.e. PC, PE, TG) in one chromatographic run. Each sample was run consecutively in positive and negative polarity. Additionally, we used data dependent acquisition (DDA) MS2 for each sample in both polarities to match the data against publicly available lipidomics databases (i.e. LipidBlast). For peak picking, peak alignment, and peak deconvolution, we used Compound Discoverer software (Thermo Scientific), and a table containing compounds with accurate mass (from positive and negative polarity) and specific retention time was generated. For data analysis, we used recently developed software called LipiDex that allows the identification of lipids by using an accurate scoring algorithm based on MS/MS fragmentation rules and chromatographic peak deconvolution. Fragmentation rules for lipids are well-documented in literature (Kui Yang, et al. Anal. Chem. 2009. 81, 4356-4368). It is known that phosphatidylethanolamine (PE) lipids mainly exhibit neutral loss fragments of m/z 141 in positive polarity and m/z 140 in negative polarity. Additionally, phosphatidylcholine (PC) lipids exhibit a fragment ion of m/z 104 and m/z 184 in positive polarity. These unique fragmentation characteristics made it possible to obtain spectral matching scores for lipid identification; only lipids with a spectral purity higher than 75% were considered for further analysis. Thus, combining both the spectral matching and the chromatographic peak purity, LipiDex allowed the accurate identification and quantitation of co-eluted isobaric lipids.

Figure 4f is very hard to grasp. Frankly, the two pictures look entirely different and I am not sure what I am really evaluating here?

Original Figure 4f (now Figure 5f) shows pictures of matched prostate tumour biopsies before and after androgen deprivation therapy (ADT) obtained from same individuals. Samples following ADT signifies castration resistant disease. Images from patient 9, who had a positive pre/post score difference were selected to illustrate enhanced DECR1 expression in castration resistant (post-treatment) disease.

Minor

Times sign and letter x should not be interchanged

Manuscript has now been proof-read and this issue has been fixed.

Reviewer #3 (Remarks to the Author):

Minor comments:

Can the authors provide details on the specific data processing performed in Renishaw Wire and what sort of custom baseline subtraction was used?

Data processing was performed in WiRE software which enabled removal of cosmic rays (using a nearest neighbour detection) following which, the baseline was subtracted using the in-built baseline subtraction intelligent fitting function (with an 11th order polynomial fitting and noise tolerance set to 1.50 which was applied to the whole spectral dataset). These details have now been added to the Materials and Methods section.

What was the actual laser power on the sample? The laser offers 500 mW (which is relatively high compared to most studies) and 100% laser power was used.

The laser power was measured as approx. 30 mW at the objective using a 60x water immersion lens with 100% laser power output from the source.

It would be good for non-experts to know why Raman is a tool of choice for this specific application. i.e., the authors should mention the high sensitivity to lipids.

We have now modified the text accordingly: “Raman spectroscopy is a non-destructive label free analytical technique that can provide rich molecularly specific information without isolation of the specific molecules under investigation. Mixtures of molecules can be identified from their specific vibrations with lipids in particular having a high Raman cross section making them ideal candidates for study by this technique.”

On page 12-13 the authors should either acquire a Raman spectrum from cholesterol or provide a citation for cholesterol Raman spectra and lipids in general.

We have now cited the appropriate reference: Tissue characterization using high wave number Raman spectroscopy

Senada Koljenovic; Tom C. Bakker Schut; Rolf Wolthuis; B. de Jong; L. Santos; Peter J. Caspers; Johan M. Kros; Gerwin J. Puppels

J. Biomedical Optics, 10(3), 031116 (2005).

REVIEWERS' COMMENTS:

Reviewer #1 (Remarks to the Author):

The authors have clarified most of the points raised and have substantially improved the manuscript. Importantly, they have now clearly demonstrated that cell death induced by DECR1 silencing indeed involves ferroptosis. Also, the additional analysis of different lipid species in cells and tumour tissue now strongly supports the conclusions of the manuscript. However, there are two minor issues that still could still be resolved/clarified.

1) The authors now also investigated phospho-ACC to demonstrate activation of AMPK. However, they also need to show pan-ACC to confirm that the increase in signal is indeed caused by increased phosphorylation.

2) The added lipidomics data clearly improve the manuscript. The new data show changes in the cellular lipidome. The authors conclude that the reduction in the amount of mono-unsaturated fatty acids (MUFAs) and an increase in poly-unsaturated fatty acids (PUFA) could be involved in enhanced ferroptosis sensitivity in DECR1 KO cells (Figure 6E and F and text on page 13). However, this analysis clearly shows that the abundance of cis-9 18:1 (oleic acid) and cis-8 18:1 is not reduced. These two fatty acids make up the vast majority of MUFA in cells, as shown in the data. As both oleic acid and palmitoleic acid have been shown to prevent ferroptosis (Magtanong et al, also cited in the manuscript), it seems surprising that the observed (25-50%) decrease in two minor MUFA species (sapienate and palmitoleate) is responsible for increased ferroptosis sensitivity. Also, there is no significant increase in arachidonic acid in one of the DECR1 KO cell lines (as suggested by the text). While the data are overall supporting the conclusions drawn, the authors should discuss this aspect more carefully in the text.

Reviewer #2 (Remarks to the Author):

Although the authors have largely resolved my concerns I have to insist on a better experimental description.

It is good to see that the authors validated their findings using GCMS quantification of FA. However, the method is not described, nor can I find a brief description in the method section, nor a related reference, nor a link to a supplementary method. Please specify this, at least citing an adequate reference how this was done.

Unfortunately the same holds true for lipidomics analysis. Its totally fine to do relative quantification and group comparison. However, as the authors state themselves: ...In addition, we did spike SPLASH lipidomic mixture (AVANTI) to each sample...Frankly I cant find this information in the paper. Sorry, but please write up the experimental part in a way so it can be repeated by others, if such details are missing that will make your work irreproducible and that would be a pity for the authors themselves and not only for others.

All conceptual concerns have satisfactorily been resolved.
Martin Giera

Reviewer #3 (Remarks to the Author):

The Raman spectroscopy queries have been fully addressed by the authors.

REVIEWERS' COMMENTS:

Reviewer #1 (Remarks to the Author):

The authors have clarified most of the points raised and have substantially improved the manuscript. Importantly, they have now clearly demonstrated that cell death induced by DECR1 silencing indeed involves ferroptosis. Also, the additional analysis of different lipid species in cells and tumour tissue now strongly supports the conclusions of the manuscript.

We are pleased to hear that the new set of experiments helped to clarify Reviewer 1's questions. We agree that all the suggested experiments have strongly contributed in improving the original manuscript.

However, there are two minor issues that still could still be resolved/clarified.

1) The authors now also investigated phospho-ACC to demonstrate activation of AMPK. However, they also need to show pan-ACC to confirm that the increase in signal is indeed caused by increased phosphorylation.

*We would like to draw the Reviewer's attention on the fact that the requested western blot (total ACC expression in ARI-resistant cell line) is depicted in **Fig. 3b**. While ACC expression level is increased in ARI-resistant cells when compared to WT LNCaP, this increase in expression is not as strong as the increase in ACC-phosphorylation that we observe in **Suppl. Fig. 4a**. Because ACC itself can be directly regulated by AR (**Fig. 4b**), we cannot be certain that the increase in ACC-phosphorylation is only due to increased AMPK activity. Therefore we referred to "increased AMPK α expression and activity" in the result section of this manuscript (page 8). Nevertheless, we have now decided to modify another sentence in the discussion from "increased AMPK activation" to "increased AMPK expression" (page 16).*

2) The added lipidomics data clearly improve the manuscript. The new data show changes in the cellular lipidome. The authors conclude that the reduction in the amount of mono-unsaturated fatty acids (MUFAs) and an increase in poly-unsaturated fatty acids (PUFA) could be involved in enhanced ferroptosis sensitivity in DECR1 KO cells (Figure 6E and F and text on page 13). However, this analysis clearly shows that the abundance of cis-9 18:1 (oleic acid) and cis-8 18:1 is not reduced. These two fatty acids make up the vast majority of MUFA in cells, as shown in the data. As both oleic acid and palmitoleic acid have been shown to prevent ferroptosis (Magtanong et al, also cited in the manuscript), it seems surprising that the observed (25-50%) decrease in two minor MUFA species (sapienate and palmitoleate) is responsible for increased ferroptosis sensitivity. Also, there is no significant increase in arachidonic acid in one of the DECR1 KO cell lines (as suggested by the text). While the data are overall supporting the conclusions drawn, the authors should discuss this aspect more carefully in the text.

We thank the Reviewer for finding this new set of data of interest. We agree that changes in the level of free arachidonic acid, but not total, did not reach significance for DECR1 KO 1 and we have now modified the main text to be more accurate. We also agree that we did not observe any change in levels of the abundant 18:1 fatty acid. While we cannot fully explain this result, our hypothesis is that DECR1, due to its enzymatic activity, will primarily affect PUFA homeostasis. Therefore, changes in

PUFA composition, rather than MUFAs, might contribute to a greater extent to the effects that we observed in the DECRI KO cells. We have now discussed this aspect in the Discussion section.

Reviewer #2 (Remarks to the Author):

Although the authors have largely resolved my concerns I have to insist on a better experimental description.

We are happy to hear that the results of the suggested experiments have contributed to resolve Reviewer 2's concerns. We have now modified our Methods section in order to improve experimental description.

It is good to see that the authors validated their findings using GCMS quantification of FA. However, the method is not described, nor can I find a brief description in the method section, nor a related reference, nor a link to a supplementary method. Please specify this, at least citing an adequate reference how this was done.

We apologize for the apparent confusion: the detailed protocol for the GCMS quantification of FA was given in the Supplementary Information section. However, given the importance of this experiment (as well as the journal's guidelines), we have now included that protocol description in the main Methods section.

Unfortunately the same holds true for lipidomics analysis. Its totally fine to do relative quantification and group comparison. However, as the authors state themselves: ...In addition, we did spike SPLASH lipidomic mixture (AVANTI) to each sample...Frankly I cant find this information in the paper. Sorry, but please write up the experimental part in a way so it can be repeated by others, if such details are missing that will make your work irreproducible and that would be a pity for the authors themselves and not only for others.

Indeed, we realized that we forgot to mention the addition of an internal standard in the original method description, and we apologize for this mistake. We have now updated the details of the method and added this information to the relevant section.

All conceptual concerns have satisfactorily been resolved.

Martin Giera

Reviewer #3 (Remarks to the Author):

The Raman spectroscopy queries have been fully addressed by the authors.

We are pleased to hear that we have positively addressed the comments from Reviewer 3.